# Accelerated Over-Relaxation Heavy-Ball Method: Achieving Global Accelerated Convergence with Broad Generalization

**Jingrong Wei**
Department of Mathematics
University of California, Irvine
Irvine, CA 92697
jingronw@uci.edu

**Long Chen** *
Department of Mathematics
University of California, Irvine
Irvine, CA 92697
chenlong@math.uci.edu

## ABSTRACT

The heavy-ball momentum method accelerates gradient descent with a momentum term but lacks accelerated convergence for general smooth strongly convex problems. This work introduces the Accelerated Over-Relaxation Heavy-Ball (AOR-HB) method, the first variant with provable global and accelerated convergence for such problems. AOR-HB closes a long-standing theoretical gap, extends to composite convex optimization and min-max problems, and achieves optimal complexity bounds. It offers three key advantages: (1) broad generalization ability, (2) potential to reshape acceleration techniques, and (3) conceptual clarity and elegance compared to existing methods.

## 1 INTRODUCTION

We first consider the convex optimization problem:

$$\min_{x \in \mathbb{R}^d} f(x), \tag{1}$$

where the objective function $f$ is $\mu$-strongly convex and $L$-smooth. Later on we consider extension to composite convex optimization $\min_x f(x) + g(x)$ with a non-smooth function $g$, and a class of min-max problems $\min_{u \in \mathbb{R}^m} \max_{p \in \mathbb{R}^n} f(u) - g(p) + \langle Bu, p \rangle$ with bilinear coupling.

**Notation.** $\mathbb{R}^d$ is $d$-dimensional Euclidean space with standard $\ell_2$-inner product $\langle \cdot, \cdot \rangle$ and the induced norm $\| \cdot \|$. $f : \mathbb{R}^d \to \mathbb{R}$ is a differentiable function. We say $f$ is $\mu$-strongly convex function when there exists $\mu > 0$ such that

$$f(y) - f(x) - \langle \nabla f(x), y - x \rangle \geq \frac{\mu}{2} \|x - y\|^2, \quad \forall\, x, y \in \mathbb{R}^d.$$

We say $f$ is $L$-smooth with $L > 0$ if its gradient is Lipschitz continuous:

$$\|\nabla f(x) - \nabla f(y)\| \leq L \|x - y\|, \quad \forall\, x, y \in \mathbb{R}^d.$$

The condition number $\kappa$ is defined as $\kappa = L/\mu$.

When $f$ is $\mu$-strongly convex and $L$-smooth, the optimization problem (1) has a unique global minimizer $x^*$. We focus on iterative methods to find $x^*$. A useful measure of convergence is the Bregman divergence of $f$, defined as $D_f(y, x) := f(y) - f(x) - \langle \nabla f(x), y - x \rangle$. In particular, $D_f(x, x^*) = f(x) - f(x^*)$, since $\nabla f(x^*) = 0$. Various bounds and identities on the Bregman divergence can be found in Appendix A.

**Heavy-ball methods and flow.** Over the past two decades, first-order methods, which rely solely on gradient information rather than the Hessian as required by Newton's method, have gained significant interest due to their efficiency and adaptability to large-scale data-driven applications and

---
*Corresponding author.

machine learning tasks (Bottou et al., 2018). Among these methods, the gradient descent method is the most straightforward and well-established algorithms. However, for ill-conditioned problems, where the condition number $\kappa \gg 1$, the gradient descent method suffers from slow convergence.

In order to accelerate the gradient descent methods, a momentum term was introduced, encouraging the method to move along search directions that utilize not only current but also previously seen information. The heavy-ball (HB) method (also known as the momentum method (Polyak, 1964) was in the form of:

$$x_{k+1} = x_k - \gamma \nabla f(x_k) + \beta(x_k - x_{k-1}), \tag{2}$$

where $\beta$ and $\gamma$ are constant parameters. Polyak motivated the method by an analogy to a "heavy ball" moving in a potential well defined by the objective function $f$. The corresponding ordinary differential equation (ODE) model is commonly referred to as the heavy-ball flow (Polyak, 1964):

$$x'' + \theta x' + \eta \nabla f(x) = 0, \tag{3}$$

where $x = x(t)$, $x'$ is taking the derivative of $t$, and $\theta$ and $\eta$ are positive constant parameters.

**Non-convergence and non-acceleration of the HB method.** Polyak (1964) showed that for (2) with $\beta = \left(\frac{\sqrt{L} - \sqrt{\mu}}{\sqrt{L} + \sqrt{\mu}}\right)^2$, $\gamma = \frac{4}{(\sqrt{L} + \sqrt{\mu})^2}$, and when $x_k$ is sufficiently close to the optimal solution $x^*$, $\|x_k - x^*\|$ converges at rate $\frac{1 - \sqrt{\rho}}{1 + \sqrt{\rho}}$, where $\rho = 1/\kappa$. Polyak's choice relies on the spectral analysis of the linear system and thus the accelerated rate is only limited to the convex and quadratic objectives and is local for iterates near $x^*$. Indeed Lessard et al. (2016) designed a non-convergent example to show that the Polyak's choice of parameters does not guarantee the global convergence for general strongly convex optimization. By changing the parameters, in Ghadimi et al. (2015); Sun et al. (2019); Saab Jr et al. (2022); Shi et al. (2022), the global linear convergence of the HB method has been established. However, the best rate is $1 - \mathcal{O}(\rho)$ given by Shi et al. (2022), which coincides with that of the gradient descent and not the accelerated rate $1 - \mathcal{O}(\sqrt{\rho})$.

The absence of acceleration is not due to a technical difficulty in the convergence analysis. Recently, Goujaud et al. (2023) have demonstrated that the HB method provably fails to achieve an accelerated convergence rate for smooth and strongly convex problems. Specifically, for any positive parameters $\beta$ and $\gamma$ in (2), either there exists an $L$-smooth, $\mu$-strongly convex function, and an initialization such that HB fails to converge; or even in the class of smooth and strongly convex quadratic function $f$, the convergence rate is not accelerated: $1 - \mathcal{O}(\rho)$. For more related works on HB methods, see Appendix B.1.

**Accelerated first-order methods.** To accelerate the HB method, one can introduce either an additional gradient step with adequate decay or an extrapolation step into the algorithm; see Wilson et al. (2021); Siegel (2019); Chen & Luo (2021).

Nesterov accelerated gradient (NAG) method (Nesterov, 1983, page 81) can be viewed as an alternative enhancement of the HB method. Nesterov's approach calculates the gradient at points that are extrapolated based on the inertial force:

$$x_{k+1} = x_k + \beta(x_k - x_{k-1}) - \gamma \nabla f(x_k + \beta(x_k - x_{k-1})), \text{ with } \beta = \frac{\sqrt{L} - \sqrt{\mu}}{\sqrt{L} + \sqrt{\mu}}, \gamma = \frac{1}{L}. \tag{4}$$

Nesterov devised the method of estimate sequences (Nesterov, 2013) to prove that (4) achieves the accelerated linear convergence rate $1 - \sqrt{\rho}$.

Later on, numerous accelerated gradient methods have been developed for smooth strongly convex optimization problems (Lin et al., 2015; Drusvyatskiy et al., 2018; Bubeck et al., 2015; Aujol et al., 2022; Van Scoy et al., 2017; Cyrus et al., 2018); to name just a few. However, little is known beyond convex optimization. One reason is that the techniques developed are often specialized on the convexity of the objective function, making them difficult to extend to non-convex cases. In machine learning terms, these approaches lack generalization ability.

**Main contributions for smooth strongly convex optimization.** We propose a variant of the HB method in the form of

$$x_{k+1} = x_k - \gamma(2\nabla f(x_k) - \nabla f(x_{k-1})) + \beta(x_k - x_{k-1}),$$
$$\gamma = \frac{1}{(\sqrt{L} + \sqrt{\mu})^2}, \qquad \beta = \frac{L}{(\sqrt{L} + \sqrt{\mu})^2}. \tag{5}$$

The most notable yet simple change is using $2\nabla f(x_k) - \nabla f(x_{k-1})$ not $\nabla f(x_k)$ to approximate $\nabla f(x)$. Namely an over-relaxation technique (Hadjidimos, 1978) is applied to the gradient term. Therefore, we name (5) accelerated over-relaxation heavy-ball (AOR-HB) method.

Rather than second-order ODEs, we consider a first-order ODE system proposed in Chen & Luo (2021):

$$x' = y - x, \quad y' = x - y - \frac{1}{\mu}\nabla f(x), \tag{6}$$

which is a special case of the HB flow (3) with $\theta = 2$ and $\eta = \frac{1}{\mu}$ when $y$ is eliminated. Although (6) is mathematically equivalent to an HB flow, the structure of this $2 \times 2$ first-order ODE system is crucial for convergence analysis and algorithmic design. The same second-order ODE can correspond to different first-order systems. For example, another one equivalent to the HB flow (3) is

$$x' = v, \quad v' = -\theta v - \eta\nabla f(x), \tag{7}$$

where $v$ represents velocity, offering a physical interpretation. However, the convergence analysis is less transparent in the form (7) or the original second-order ODE (3).

We obtain AOR-HB by discretization of (6) with two iterates $(x_k, y_k)$ cf. (14). The AOR is used to symmetrize the error equation (19), which represents a novel contribution compared to Luo & Chen (2022) and Chen & Luo (2021). The choice of parameters $\beta$ and $\gamma$ in (5) is derived from the time step-size $\alpha = \sqrt{\mu/L}$. We rigorously prove that AOR-HB enjoys the global linear convergence with accelerated rate, which closes a long-standing theoretical gap in optimization theory.

**THEOREM 1.1** (Convergence of AOR-HB method). *Suppose $f$ is $\mu$-strongly convex and $L$-smooth. Let $(x_k, y_k)$ be generated by scheme (14) with initial value $(x_0, y_0)$ and step size $\alpha = \sqrt{\mu/L}$. Then there exists a constant $C_0 = C_0(x_0, y_0, \mu, L)$ so that we have the accelerated linear convergence*

$$f(x_{k+1}) - f(x^*) + \frac{\mu}{2}\|y_{k+1} - x^*\|^2 \le C_0 \left(\frac{1}{1 + \frac{1}{2}\sqrt{\mu/L}}\right)^k, \quad k \ge 1. \tag{8}$$

**Remark on non-strongly convex optimization.** When $\mu = 0$, we propose a variant of the AOR-HB method that incorporates the dynamic time rescaling introduced in (Luo & Chen, 2022):

$$x_{k+1} = x_k - \frac{k}{k+3}\frac{1}{L}\left(2\nabla f(x_k) - \nabla f(x_{k-1})\right) + \frac{k}{k+3}(x_k - x_{k-1}), \tag{9}$$

which achieves an accelerated rate of $\mathcal{O}(1/k^2)$, comparable to NAG. We refer the details and the proofs to Appendix D. Another approach to extend the acceleration involves using the perturbed objective $f(x) + \frac{\epsilon}{2}\|x\|^2$, as discussed in (Lessard et al., 2016, Section 5.4).

**Relation to other acceleration methods.** The AOR term $2\nabla f(x_k) - \nabla f(x_{k-1})$ can be treated as adding a gradient correction in the high resolution ODE model (Shi et al., 2022). AOR-HB (5) can be also rendered as a special case of the 'SIE' iteration described in Zhang et al. (2021). However, the parameter choice ($\sqrt{s} = \frac{1}{L}$ and $m = 1$) recovering AOR-HB, does not satisfy the condition in their convergence analysis (Zhang et al., 2021, Theorem 3).

For strongly convex optimization, acceleration can be viewed from various perspectives, with the resulting three-term formulas differing only by a higher-order $\mathcal{O}(\rho)$ perturbation. With a proper change of variables, NAG (4) can be seen as a parameter perturbation of the AOR-HB in the form of (5). The performance of AOR-HB are thus comparable to NAG but less precise than triple momentum (TM) methods (Van Scoy et al., 2017) for strongly convex optimization.

However, extending these methods (NAG, TM, SIE, and high-resolution ODEs) beyond the convex optimization is rare and challenging. In contrast, AOR-HB has a superior generalization capability as the true driving force of acceleration is better captured by the $2 \times 2$ first-order ODE model (6).

**Extension to composite convex minimization.** Consider the optimization problem:

$$\min_x f(x) + g(x), \tag{10}$$

where $f$ is $\mu$-strongly convex and $L$-smooth, and $g$ is a convex but maybe non-smooth function. To handle this, we only need to split the gradient term in (6) into $\nabla f(x) + \partial g(y)$ and use an implicit discretization scheme for $y$. This is equivalent to applying the proximal operator

$$\mathrm{prox}_{\lambda g}(x) := \min_y g(y) + \frac{1}{2\lambda}\|y - x\|^2,$$

which we assume is available. The convergence analysis of the AOR-HB-composite Algorithm 2 directly follows from Theorem 1.1. Numerically, Algorithm 2 also performs well for certain non-smooth and non-convex functions $g$ when its proximal operator is available, further demonstrating the generalization capability of the AOR-HB method.

**Extension to a class of saddle point problems.** We extend the AOR-HB method (Algorithm 3) to a strongly-convex-strongly-concave saddle point system with bilinear coupling, defined as follows:

$$\min_{u \in \mathbb{R}^m} \max_{p \in \mathbb{R}^n} \mathcal{L}(u, p) := f(u) - g(p) + \langle Bu, p \rangle \tag{11}$$

where $B \in \mathbb{R}^{n \times m}$ is a matrix and $B^\top$ denotes its transpose, $f : \mathbb{R}^m \to \mathbb{R}$ and $g : \mathbb{R}^n \to \mathbb{R}$ are strongly convex functions with convexity constant $\mu_f$ and $\mu_g$, respectively. Problem (11) has a large number of applications, some of which we briefly introduce in Appendix B.4.

Only few optimal first-order algorithms for saddle point problems have been developed recently (Metelev et al., 2024; Thekumparampil et al., 2022; Kovalev et al., 2022; Jin et al., 2022). In particular, Thekumparampil et al. (2022) introduced the Lifted Primal-Dual (LPD) method, the first optimal algorithm for problem (11). However, LPD involves five parameters, whereas AOR-HB-saddle requires only two parameters: $\mu$ and $L$. Numerically, AOR-HB-saddle achieves highly efficient performance while retaining the simplicity of the HB method. We state the convergence result below and refer to Appendix G.3 for a proof.

**THEOREM 1.2** (Convergence of AOR-HB-saddle method). *Suppose $f$ is $\mu_f$-strongly convex and $L_f$-smooth, $g$ is $\mu_g$-strongly convex and $L_g$-smooth. Let $(u_k, v_k, p_k, q_k)$ be generated by Algorithm 3 with initial value $(u_0, v_0, p_0, q_0)$ and step size $\alpha = \max_{\beta \in (0,1)} \min\left\{\sqrt{\beta}\min\left\{\sqrt{\frac{\mu_f}{L_f}}, \sqrt{\frac{\mu_g}{L_g}}\right\}, (1-\beta)\frac{\sqrt{\mu_f \mu_g}}{\|B\|}\right\}$. Then there exists a non-negative constant $C_0 = C_0(u_0, v_0, p_0, q_0, \mu_f, L_f, \mu_g, L_g)$ so that we have the linear convergence*

$$D_f(u_{k+1}, u^*) + D_g(p_{k+1}, p^*) + \frac{\mu_f}{2}\|v_{k+1} - u^*\|^2 + \frac{\mu_g}{2}\|q_{k+1} - p^*\|^2 \leq C_0 \left(\frac{1}{1 + \alpha/2}\right)^k.$$

**Extension to a class of monotone operator equations.** Our approach provides a unified framework for convex optimization, saddle-point problems, and monotone operator equations. Consider

$$\begin{pmatrix} x' \\ y' \end{pmatrix} = \begin{pmatrix} -I & I \\ I - \mu^{-1}A & -I - \mu^{-1}N \end{pmatrix} \begin{pmatrix} x \\ y \end{pmatrix}.$$

- Strongly convex optimization: $A(x) = \nabla f(x)$, $N = 0$. This is the accelerated gradient flow developed in Luo & Chen (2022) and Chen & Luo (2021).
- Composite convex optimization: $A(x) = \nabla f(x)$, $N(y) = \partial g(y)$.
- Saddle-point problem with bilinear coupling: $A(x) = \begin{pmatrix} \nabla f(u) & 0 \\ 0 & \nabla g(p) \end{pmatrix}$, $N = \begin{pmatrix} 0 & B^\top \\ B & 0 \end{pmatrix}$.
- A class of monotone operators: $A(x) = \nabla F(x)$, $N$ is linear and skew-symmetric.

In the discretization, AOR can be utilized to faithfully preserve the structure of the flow. The AOR term is also connected to the extra-gradient methods originally proposed by (Korpelevich, 1976; Popov, 1980) for saddle point problems, which has only non-accelerated rate $\mathcal{O}(\kappa|\log \epsilon|)$. We refer to Appendix B.3 for more discussion.

## 2 AOR-HB METHOD FOR CONVEX OPTIMIZATION

**Strong Lyapunov property.** To simplify notation, introduce $\mathbf{z} = (x, y)^\top$ and $\mathbf{z}^* = (x^*, x^*)^\top$. Let $\mathcal{G}(\mathbf{z})$ be a vector field with $\mathcal{G}(\mathbf{z}^*) = 0$. For a generic ODE $\mathbf{z}' = \mathcal{G}(\mathbf{z})$, a Lyapunov function is a non-negative function $\mathcal{E}(\mathbf{z})$ satisfying $-\langle \nabla \mathcal{E}(\mathbf{z}), \mathcal{G}(\mathbf{z}) \rangle \geq 0$ for all $\mathbf{z}$ near $\mathbf{z}^*$ and $\mathcal{E}(\mathbf{z}) = 0$ if and only if $\mathbf{z} = \mathbf{z}^*$. In order to get the exponential stability, Chen & Luo (2021) introduce the so-called strong Lyapunov property: there exists $c > 0$ such that

$$- \langle \nabla \mathcal{E}(\mathbf{z}), \mathcal{G}(\mathbf{z}) \rangle \geq c \, \mathcal{E}(\mathbf{z}) \quad \forall \mathbf{z} \in \mathbb{R}^d. \tag{12}$$

**LEMMA 2.1.** *Let $\mathbf{z}' = \mathcal{G}(\mathbf{z})$ with $\mathbf{z}(0) = \mathbf{z}_0$. Assume there exists a Lyapunov function $\mathcal{E}(\mathbf{z})$ satisfying the strong Lyapunov property (12). Then we have the exponential stability*

$$\mathcal{E}(\mathbf{z}) \leq e^{-ct} \mathcal{E}(\mathbf{z}_0).$$

*Proof.* By the chain rule and the strong Lyapunov property:

$$\frac{\mathrm{d}}{\mathrm{d}t} \mathcal{E}\left(\mathbf{z}(t)\right) = \langle \nabla \mathcal{E}(\mathbf{z}), \mathbf{z}' \rangle = \langle \nabla \mathcal{E}(\mathbf{z}), \mathcal{G}(\mathbf{z}) \rangle \leq -c \mathcal{E}(\mathbf{z}(t)), \tag{13}$$

and the exponential stability follows. $\qquad\square$

For the system (6), let $\mathcal{E}(x, y) = f(x) - f(x^*) + \frac{\mu}{2} \|y - x^*\|^2$. By direct calculation and the $\mu$-convexity of $f$, we obtain the strong Lyapunov property

$$- \langle \nabla \mathcal{E}(x, y), \mathcal{G}(x, y) \rangle = \langle \nabla f(x), x - x^* \rangle - \mu \langle x - y, y - x^* \rangle$$

$$= \langle \nabla f(x), x - x^* \rangle - \frac{\mu}{2} \left( \|x - x^*\|^2 - \|x - y\|^2 - \|y - x^*\|^2 \right)$$

$$\geq f(x) - f(x^*) + \frac{\mu}{2} \|y - x^*\|^2 + \frac{\mu}{2} \|x - y\|^2 = \mathcal{E}(x, y) + \frac{\mu}{2} \|x - y\|^2.$$

As shown above, the strong Lyapunov property and exponential stability can be established more straightforwardly using the first-order ODE system (6) rather than the original second-order ODE (3) or its equivalent form (7).

**AOR-HB method for smooth strongly convex minimization problems.** Based on discretization of (6), we propose an Accelerated Over-Relaxation Heavy-Ball (AOR-HB) method:

$$\frac{x_{k+1} - x_k}{\alpha} = y_k - x_{k+1}, \tag{14a}$$

$$\frac{y_{k+1} - y_k}{\alpha} = x_{k+1} - y_{k+1} - \frac{1}{\mu}(2\nabla f(x_{k+1}) - \nabla f(x_k)). \tag{14b}$$

Setting $\alpha = \sqrt{\mu/L}$ and eliminating $y_k$ leads to the formulation (5), which is summarized in Algorithm 1. The gradient $\nabla f(x_{k+1})$ can be reused in the next iteration, so essentially only one gradient evaluation is required per iteration in Algorithm 1.

---

**Algorithm 1** Accelerated Over-Relaxation Heavy-Ball Method (AOR-HB)

---

1: **Parameters:** $x_0, x_1 \in \mathbb{R}^d, L, \mu$. **Set** $\gamma = \frac{1}{(\sqrt{L} + \sqrt{\mu})^2}, \beta = \frac{L}{(\sqrt{L} + \sqrt{\mu})^2}$.
2: **for** $k = 1, 2, \ldots$ **do**
3: $\quad x_{k+1} = x_k - \gamma(2\nabla f(x_k) - \nabla f(x_{k-1})) + \beta(x_k - x_{k-1})$
4: **end for**
5: **return** $x_{k+1}$

---

The convergence rate in Theorem 1.1 is global and accelerated, in the sense that to obtain $f(x_k) - f(x^*) \leq \epsilon$ and $\|x_k - x^*\| \leq \epsilon$, we need at most $\mathcal{O}\left(\sqrt{\kappa} |\log \epsilon|\right)$ iterations. The iteration complexity is optimal for first-order methods (Nesterov, 2013). We give a proof for Theorem 1.1 in Appendix C and will outline the key steps using convex quadratic objectives here.

**Preliminaries for convergence analysis.** For a symmetric matrix $A$, introduce $\langle x, y \rangle_A = \langle Ax, y \rangle = \langle x, Ay \rangle$ and $\| \cdot \|_A^2 := \langle \cdot, \cdot \rangle_A$. Consider the quadratic and convex function $f(x) = \frac{1}{2}\|x\|_A^2 - \langle b, x \rangle + c$ for $b \in \mathbb{R}^d, c \in \mathbb{R}$ and symmetric and positive definite (SPD) matrix $A$ with bound $0 < \mu \leq \lambda(A) \leq L$, where $\lambda(A)$ denotes a generic eigenvalue of matrix $A$, and $\lambda_{\min}(A)$ and $\lambda_{\max}(A)$ are minimal and maximal eigenvalues of $A$, respectively.

For two symmetric matrices $D, A$, we denote $A \prec D$ if $D - A$ is SPD and $A \preceq D$ if $D - A$ is positive semi-definite. One can easily verify that $A \preceq D$ is equivalent to $\lambda_{\max}(D^{-1}A) \leq 1$ when $D$ is non-singular or $\lambda_{\min}(A^{-1}D) \geq 1$ when $A$ is non-singular.

Denote by $\mathcal{D} = \begin{pmatrix} A & 0 \\ 0 & \mu I \end{pmatrix}$ and $\mathcal{A}^{\mathrm{sym}} = \begin{pmatrix} 0 & A \\ A & 0 \end{pmatrix}$. Define two Lyapunov functions

$$\mathcal{E}(\mathbf{z}) := \frac{1}{2}\|\mathbf{z} - \mathbf{z}^*\|_{\mathcal{D}}^2, \quad \text{and} \quad \mathcal{E}^\alpha(\mathbf{z}) := \frac{1}{2}\|\mathbf{z} - \mathbf{z}^*\|_{\mathcal{D}+\alpha\mathcal{A}^{\mathrm{sym}}}^2.$$

By direct calculation, we have $\lambda(\mathcal{D}^{-1}\mathcal{A}^{\mathrm{sym}}) = \pm\sqrt{\lambda(A)/\mu}$ which implies

$$(1 - \alpha/\sqrt{\rho})\mathcal{D} \preceq \mathcal{D} + \alpha\mathcal{A}^{\mathrm{sym}} \preceq (1 + \alpha/\sqrt{\rho})\mathcal{D}, \qquad \rho = \mu/L. \tag{15}$$

Therefore for $0 \leq \alpha \leq \sqrt{\rho}$,

$$0 \leq (1 - \alpha/\sqrt{\rho})\mathcal{E}(\mathbf{z}) \leq \mathcal{E}^\alpha(\mathbf{z}) \leq 2\mathcal{E}(\mathbf{z}), \quad \forall\, \mathbf{z} = (x, y)^\top \in \mathbb{R}^{2d}. \tag{16}$$

**Convergence analysis for quadratic functions.** The AOR-HB method (14) can be written as a correction of the implicit Euler discretization of the flow (6)

$$\mathbf{z}_{k+1} - \mathbf{z}_k = \alpha\mathcal{G}(\mathbf{z}_{k+1}) - \alpha \begin{pmatrix} 0 & I \\ \frac{1}{\mu}A & 0 \end{pmatrix} (\mathbf{z}_{k+1} - \mathbf{z}_k). \tag{17}$$

As $\mathcal{E}$ is quadratic, $D_{\mathcal{E}}(\mathbf{z}, \mathbf{y}) = \frac{1}{2}\|\mathbf{z} - \mathbf{y}\|_{\mathcal{D}}^2$. So use the definition of Bregman divergence, we have the identity for the difference of the Lyapunov function $\mathcal{E}$ at consecutive steps:

$$\mathcal{E}(\mathbf{z}_{k+1}) - \mathcal{E}(\mathbf{z}_k) = \langle \nabla\mathcal{E}(\mathbf{z}_{k+1}), \mathbf{z}_{k+1} - \mathbf{z}_k \rangle - \frac{1}{2}\|\mathbf{z}_k - \mathbf{z}_{k+1}\|_{\mathcal{D}}^2. \tag{18}$$

Substitute (17) into (18) and expand the cross term as:

$$\langle \nabla\mathcal{E}(\mathbf{z}_{k+1}), \mathbf{z}_{k+1} - \mathbf{z}_k \rangle = \alpha\langle \nabla\mathcal{E}(\mathbf{z}_{k+1}), \mathcal{G}(\mathbf{z}_{k+1}) \rangle - \alpha\langle \mathbf{z}_{k+1} - \mathbf{z}^*, \mathbf{z}_{k+1} - \mathbf{z}_k \rangle_{\mathcal{A}^{\mathrm{sym}}}. \tag{19}$$

The AOR $Ax \approx 2Ax_{k+1} - Ax_k = Ax_{k+1} + A(x_{k+1} - x_k)$ is used to make the last term in (19) associated to a symmetric matrix $\mathcal{A}^{\mathrm{sym}}$. With such symmetrization, we can use identity of squares

$$\langle \mathbf{z}_{k+1} - \mathbf{z}^*, \mathbf{z}_{k+1} - \mathbf{z}_k \rangle_{\mathcal{A}^{\mathrm{sym}}} = \frac{1}{2}\left(\|\mathbf{z}_{k+1} - \mathbf{z}^*\|_{\mathcal{A}^{\mathrm{sym}}}^2 + \|\mathbf{z}_{k+1} - \mathbf{z}_k\|_{\mathcal{A}^{\mathrm{sym}}}^2 - \|\mathbf{z}_k - \mathbf{z}^*\|_{\mathcal{A}^{\mathrm{sym}}}^2\right).$$

Substitute back to (18) and rearrange terms, we obtain the following identity:

$$\mathcal{E}^\alpha(\mathbf{z}_{k+1}) - \mathcal{E}^\alpha(\mathbf{z}_k) = \alpha\langle \nabla\mathcal{E}(\mathbf{z}_{k+1}), \mathcal{G}(\mathbf{z}_{k+1}) \rangle - \frac{1}{2}\|\mathbf{z}_{k+1} - \mathbf{z}_k\|_{\mathcal{D}+\alpha\mathcal{A}^{\mathrm{sym}}}^2, \tag{20}$$

which holds for arbitrary step size $\alpha$.

Now take $\alpha = \sqrt{\rho}$. Due to (15), the last term in (20) is non-positive. Use the strong Lyapunov property $-\langle \nabla\mathcal{E}(\mathbf{z}), \mathcal{G}(\mathbf{z}) \rangle \geq \mathcal{E}(\mathbf{z})$, and the bound $\mathcal{E}^\alpha(\mathbf{z}) \leq 2\mathcal{E}(\mathbf{z})$ in (16) to get

$$\mathcal{E}^\alpha(\mathbf{z}_{k+1}) - \mathcal{E}^\alpha(\mathbf{z}_k) \leq -\alpha\mathcal{E}(\mathbf{z}_{k+1}) \leq -\frac{\alpha}{2}\mathcal{E}^\alpha(\mathbf{z}_{k+1}), \tag{21}$$

which implies the global linear convergence of $\mathcal{E}^\alpha$:

$$\mathcal{E}^\alpha(\mathbf{z}_{k+1}) \leq \frac{1}{1+\alpha/2}\mathcal{E}^\alpha(\mathbf{z}_k) \leq \left(\frac{1}{1+\alpha/2}\right)^{k+1}\mathcal{E}^\alpha(\mathbf{z}_0), \quad k \geq 0. \tag{22}$$

Moreover, (21) implies

$$\mathcal{E}(\mathbf{z}_{k+1}) \leq \frac{1}{\alpha}(\mathcal{E}^\alpha(\mathbf{z}_k) - \mathcal{E}^\alpha(\mathbf{z}_{k+1})) \leq \frac{1}{\alpha}\mathcal{E}^\alpha(\mathbf{z}_k) \leq \frac{1}{\alpha}\left(\frac{1}{1+\alpha/2}\right)^k\mathcal{E}^\alpha(\mathbf{z}_0),$$

which implies (8) with $C_0 = \frac{1}{\alpha}\mathcal{E}^\alpha(\mathbf{z}_0) \geq 0$. Note that $C_0 = \mathcal{O}(\sqrt{\kappa})$ is large. However, when considering the convergence of $\mathcal{E}^\alpha$ as in (22), this dependency is absent.

**AOR-HB method for composite convex optimization.** For the composite convex optimization problem (10), AOR-HB can be seamlessly extended by using:

$$x' = y - x, \quad \mu y' - [\mu(x - y) - \nabla f(x)] \in \partial g(y), \tag{23}$$

where $\partial g(\cdot)$ denotes the set of subgradient. The strong Lyapunov property and exponential stability still hold (see Appendix E) as the modification only bring an additional monotone term.

We use implicit discretization in $g$ and apply AOR to $\nabla f$:

$$\frac{x_{k+1} - x_k}{\alpha} = y_k - x_{k+1}, \quad \mu \frac{y_{k+1} - y_k}{\alpha} - \mu(x_{k+1} - y_{k+1}) + (2\nabla f(x_{k+1}) - \nabla f(x_k)) \in \partial g(y_{k+1}).$$

Given the proximal operator of $g$, an equivalent and computation-favorable formulation is proposed in Algorithm 2. In the convergence analysis, since the non-smooth part is discretized implicitly, no difference arises in the error equation (17). Therefore, the convergence rate and proof are identical to those in Theorem 1.1; see Theorem E.1. In contrast, it requires considerable effort to generalize Nesterov's accelerated gradient method to the composite case Beck & Teboulle (2009).

---

**Algorithm 2** Accelerated Over-Relaxation Heavy-Ball Method for Convex Composite Minimization (AOR-HB-composite)

---

1: **Parameters:** $x_0, y_0 \in \mathbb{R}^d, L, \mu$. **Set** $\alpha = \sqrt{\frac{\mu}{L}}$ and $\lambda = \frac{\alpha}{(1+\alpha)\mu}$.
2: **for** $k = 1, 2, \ldots$ **do**
3: $\quad x_{k+1} = \frac{1}{1+\alpha}(x_k + \alpha y_k)$
4: $\quad y_{k+1} = \text{prox}_{\lambda g}(z_k)$ where $z_k = \frac{1}{1+\alpha}(y_k + \alpha x_{k+1}) - \lambda(2\nabla f(x_{k+1}) - \nabla f(x_k))$
5: **end for**
6: **return** $y_{k+1}$

---

## 3 AOR-HB Method for a Class of Saddle Point Problems

In this section, we extend the flow (3) to min-max problems and propose an accelerated first-order method for strongly-convex-strongly-concave saddle-point problems with bilinear coupling.

The solution $(u^*, p^*)$ to the min-max problem (11) is called a saddle point of $\mathcal{L}(u, p) = f(u) - g(p) + \langle Bu, p \rangle$:

$$\mathcal{L}(u^*, p) \leq \mathcal{L}(u^*, p^*) \leq \mathcal{L}(u, p^*), \quad \forall u \in \mathbb{R}^m, p \in \mathbb{R}^n.$$

As $\mathcal{L}(\cdot, p)$ is strongly convex for any given $p$ and $\mathcal{L}(u, \cdot)$ is strongly concave for any given $u$, we called it a strongly-convex-strongly-concave saddle point problem. Under this setting, the saddle point $(u^*, p^*)$ exists uniquely and is necessarily a critical point of $\mathcal{L}(u, p)$ satisfying:

$$\nabla \mathcal{L}(u^*, p^*) := \begin{pmatrix} \nabla f & B^\top \\ B & -\nabla g \end{pmatrix} \begin{pmatrix} u^* \\ p^* \end{pmatrix} = \begin{pmatrix} 0 \\ 0 \end{pmatrix}, \tag{24}$$

where $\langle \nabla f, u \rangle := \nabla f(u)$ and $\langle \nabla g, p \rangle := \nabla g(p)$.

**HB-saddle flow.** We propose the following HB-saddle flow for solving the min-max problem (11):

$$u' = v - u, \quad v' = u - v - \frac{1}{\mu_f}(\nabla f(u) + B^\top q),$$

$$p' = q - p, \quad q' = p - q - \frac{1}{\mu_g}(\nabla g(p) - Bv). \tag{25}$$

The HB-saddle flow (25) preserves the strong Lyapunov property (see Appendix G.2) and thus the exponential stability of $(u^*, p^*)$ for (25) follows.

For the gradient term, we use $J\nabla \mathcal{L}$ instead of $\nabla \mathcal{L}$ with $J = \begin{pmatrix} I & 0 \\ 0 & -I \end{pmatrix}$. When $\nabla f$ and $\nabla g$ are linear, $\nabla \mathcal{L}$ is symmetric but not monotone. $J\nabla \mathcal{L}$ is non-symmetric but strongly monotone. Namely

$$\langle J\nabla \mathcal{L}(x) - J\nabla \mathcal{L}(y), x - y \rangle \geq \min\{\mu_f, \mu_g\}\|x - y\|^2, \quad \forall x = (u, p)^\top, y = (v, q)^\top.$$

The strong monotonicity has essential similarity to strongly convexity. Comparing to the convex optimization, one extra difficulty is the bilinear coupling $\langle Bu, p \rangle$ which can be again handled by the AOR technique in discretization.

**AOR-HB-saddle method.** We propose an Accelerated Over-Relaxation Heavy-Ball (AOR-HB-saddle) Method for solving the min-max problem (11) in Algorithm 3. Each iteration requires 2 matrix-vector products and 2 gradient evaluations if we store $\nabla f(u_k)$ and $\nabla g(p_k)$.

---

**Algorithm 3** Accelerated over-relaxation Heavy-ball method for strongly-convex-strongly-concave saddle point problems with bilinear coupling (AOR-HB-saddle)

---

1: **Parameters:** $u_0, v_0 \in \mathbb{R}^m, p_0, q_0 \in \mathbb{R}^n, L_f, L_g, \mu_f, \mu_g, \|B\|$.

2: **Set** $\alpha = \max_{\beta \in (0,1)} \min \left\{ \sqrt{\beta} \min \left\{ \sqrt{\frac{\mu_f}{L_f}}, \sqrt{\frac{\mu_g}{L_g}} \right\}, (1-\beta) \frac{\sqrt{\mu_f \mu_g}}{\|B\|} \right\}$.

3: **for** $k = 0, 1, 2, \ldots$ **do**

4: $\quad u_{k+1} = \dfrac{1}{1+\alpha}(u_k + \alpha v_k); \quad p_{k+1} = \dfrac{1}{1+\alpha}(p_k + \alpha q_k)$

5: $\quad v_{k+1} = \dfrac{1}{1+\alpha}\left[ v_k + \alpha u_{k+1} - \dfrac{\alpha}{\mu_f}(2\nabla f(u_{k+1}) - \nabla f(u_k) + B^\top q_k) \right]$

6: $\quad q_{k+1} = \dfrac{1}{1+\alpha}\left[ q_k + \alpha p_{k+1} - \dfrac{\alpha}{\mu_g}(2\nabla g(p_{k+1}) - \nabla g(p_k) - B(2v_{k+1} - v_k)) \right]$

7: **end for**

8: **return** $u_{k+1}, v_{k+1}, p_{k+1}, q_{k+1}$

---

The convergence rate in Theorem 1.2 is global and accelerated, meaning that to obtain $\|(u_k, p_k) - (u^*, p^*)\| \leq \epsilon$ and $\|(v_k, q_k) - (u^*, p^*)\| \leq \epsilon$, we need at most $\mathcal{O}\left( \sqrt{L_f/\mu_f + L_g/\mu_g + \|B\|^2/(\mu_f \mu_g)} |\log \epsilon| \right)$ iterations. The iteration complexity is optimal for first-order methods for saddle point problems (Zhang et al., 2022).

***Remark* 3.1.** We can treat the coupling implicitly: Line 5 and 6 in Algorithm 3 are replaced by

$$
\begin{aligned}
v_{k+1} &= \frac{1}{1+\alpha}\left[ v_k + \alpha u_{k+1} - \frac{\alpha}{\mu_f}(2\nabla f(u_{k+1}) - \nabla f(u_k) + B^\top q_{k+1}) \right], \\
q_{k+1} &= \frac{1}{1+\alpha}\left[ q_k + \alpha p_{k+1} - \frac{\alpha}{\mu_g}(2\nabla g(p_{k+1}) - \nabla g(p_k) - Bv_{k+1}) \right].
\end{aligned}
\tag{26}
$$

Now $(v_{k+1}, q_{k+1})$ are coupled together and can be computed by inverting $\begin{pmatrix} (1+\alpha)I & \frac{\alpha}{\mu_f}B^\top \\ -\frac{\alpha}{\mu_g}B & (1+\alpha)I \end{pmatrix}$.

It is sufficient to compute $\left( (1+\alpha)^2 I + \frac{\alpha^2}{\mu_f \mu_g} BB^\top \right)^{-1}$ or $\left( (1+\alpha)^2 I + \frac{\alpha^2}{\mu_f \mu_g} B^\top B \right)^{-1}$, whichever is a relative small size matrix and can be further replaced by an inexact inner solver. We name the method as AOR-HB-saddle-I(implicit) and the convergence rate can be improved to $1 - \mathcal{O}(\sqrt{\rho})$ with $\rho = \min \{\mu_f/L_f, \mu_g/L_g\}$; see Appendix H for the formal results.

## 4 NUMERICAL RESULTS

In this section, we evaluate the performance of our AOR-HB methods on a suite of optimization problems. All numerical experiments were conducted using MATLAB R2022a on a desktop computer equipped with an Intel Core i7-6800K CPU operating at 3.4 GHz and 32GB of RAM. We compare the results against several state-of-the-art optimization algorithms from the literature.

### 4.1 SMOOTH CONVEX MINIMIZATION

We test our AOR-HB method (Algorithm 1) and compare it with other first-order algorithms including: (i) GD: gradient descent; (ii) NAG: Nesterov acceleration (Nesterov, 2013); (iii) HB: Polyak's momentum method (Polyak, 1964); (iv) TM: triple momentum method (Van Scoy et al., 2017) and (v) ADR (Aujol et al., 2022).

First, we test the algorithms using smooth multidimensional piecewise objective functions borrowed from Van Scoy et al. (2017). Let

$$
f(x) = \sum_{i=1}^{p} h\left(a_i^\top x - b_i\right) + \frac{\mu}{2}\|x\|^2, \quad h(x) = \begin{cases} \frac{1}{2}x^2 e^{-r/x}, & x > 0 \\ 0, & x \leq 0 \end{cases},
\tag{27}
$$

where $A = [a_1, \ldots, a_p] \in \mathbb{R}^{d \times p}$ and $b \in \mathbb{R}^p$ with $\|A\| = \sqrt{L - \mu}$. Then $f$ is $\mu$- strongly convex and $L$-smooth. We randomly generate the components of $A$ and $b$ from the normal distribution and then scale $A$ so that $\|A\| = \sqrt{L - \mu}$. We set $\mu = 1, L = 10^4, d = 100, p = 5$ and $r = 10^{-6}$.

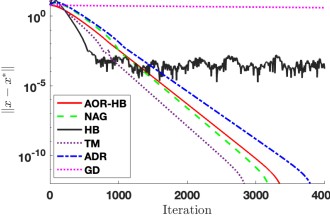

Figure 1: Simulation results for the objective function (27).

Figure 1 illustrates the logarithm of $\|x_k - x^*\|$ for each algorithm's iterates. The HB method reaches a plateau and fails to converge, while the other algorithms demonstrate global and linear convergence. As expected, GD, being non-accelerated, converges slowly due to the large condition number. The AOR-HB method performs as efficiently as NAG and ADR. The TM method achieves a convergence rate of $(1 - \sqrt{\mu/L})^2$, slightly outpacing other accelerated algorithms. However, TM requires more parameter tuning and has not yet been extended beyond strongly smooth convex optimization.

Next, we report the numerical simulations on a logistic regression problem with $\ell_2$ regularizer:

$$\min_{x \in \mathbb{R}^d} \left\{ \sum_{i=1}^{m} \log \left( 1 + \exp \left( -b_i a_i^\top x \right) \right) + \frac{\lambda}{2} \|x\|^2 \right\}, \quad (28)$$

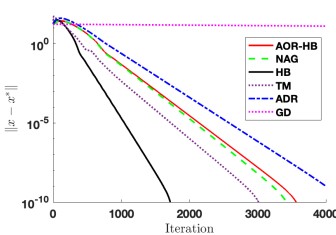

Figure 2: Simulation results for the logistic regression problem (28).

where $(a_i, b_i) \in \mathbb{R}^d \times \{-1, 1\}, i = 1, 2, \ldots, m$. For the logistic regression problem, $\mu = \lambda$ and $L = \lambda_{\max} \left( \sum_{i=1}^m a_i a_i^\top \right) + \lambda$. The data $a_i$ and $b_i$ are generated by the normal distribution and Bernoulli distribution, respectively. We set $\lambda = 0.1, d = 1000$, and $n = 50$. As illustrated by Figure 2, this is an example where HB converges and it converges fastest. However, such fast convergence lacks theoretical guarantees and may fail in cases like the one depicted in Figure 1. This highlights the importance of robust theoretical convergence analysis rather than relying on empirical success alone.

## 4.2 COMPOSITE CONVEX AND NON-CONVEX MINIMIZATION

We test the performance of AOR-HB-composite method (Algorithm 2) on non-smooth optimization problems and compare with two well-known methods: the Fast Iterative Shrinkage-Thresholding Algorithm (FISTA) Beck & Teboulle (2009) and the Accelerated Proximal Gradient (APG) algorithm proposed in Li & Lin (2015).

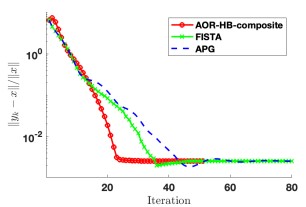

Figure 3: Comparison of $\ell_1$ minimization methods.

We first consider the Lasso problem:

$$\min_x \frac{1}{2} \|Ax - b\|_2^2 + \lambda \|x\|_1,$$

which has wide applications in compressed sensing, signal processing, and statistical learning, among other fields (Tibshirani, 1996). We generate the matrix $A$ with size $1024 \times 256$ from Gaussian random matrices and a ground-truth sparse vector $x$ of sparsity 5. The data vector obtained by matrix-vector multiplication $b = Ax$. We set $\lambda = 0.8$ and use the step size $\alpha = 1/L$ in FISTA and APG. We plot the decay of the relative $\ell_2$ error in Figure 3. All the methods cannot reach the accuracy of $10^{-3}$. This is because the ground-truth (sparse) solution may not be the minimizer of the corresponding minimization problem.

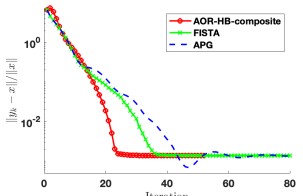

Figure 4: Comparison of $\ell_1 - \ell_2$ minimization methods.

Then we test the non-convex $\ell_1 - \ell_2$ case where the $\ell_1$ regularizer is substituted by $\|x\|_1 - \|x\|_2$. The proximal operator for $g(x) := \lambda(\|x\|_1 - \|x\|_2)$ is given in Yin et al. (2015). Using such non-smooth and non-convex function $g$ will produce solution with better sparsity. The other settings remain the same as the Lasso problem. Figure 3 and 4 show the convergence of the relative $\ell_2$ error, where all methods converges but AOR-HB outperforms others.

### 4.3 SADDLE POINT PROBLEMS

We compare AOR-HB-saddle (Algorithm 3) with the following algorithms: (i) AG-OG: accelerated gradient-optimistic gradient method with restarting regime (Li et al., 2023); (ii) APDG: accelerated primal-dual gradient method (Kovalev et al., 2022); (iii) LPD: lifted primal-dual method (Thekumparampil et al., 2022); (iv) EG: a variant of extra-gradient method (Mokhtari et al., 2020a).

We consider policy evaluation problems in reinforcement learning (Du et al., 2017) when finding minimum of the mean squared projected Bellman error (MSPBE):

$$\arg\min_u \frac{1}{2}\|Bu - b\|_{C^{-1}}^2 + \frac{1}{2}\|u\|^2.$$

The corresponding saddle point problem (see Appendix B.4) is

$$\min_{u \in \mathbb{R}^m} \max_{p \in \mathbb{R}^n} \frac{1}{2}\|u\|^2 - \frac{1}{2}\|p\|_C^2 - \langle b, p \rangle + \langle Bu, p \rangle. \tag{29}$$

The saddle point formulation saves the computation cost of inverting $C$. In this example, $\mu_f = L_f = 1$, $\mu_g = \lambda_{\min}(C)$ and $L_g = \lambda_{\max}(C)$. We generate random matrices $B$ and $C$ such that $\mu_g = 1$ and the condition number $\kappa_g = \|B\|^2$. We set $m = 2500$ and $n = 50$.

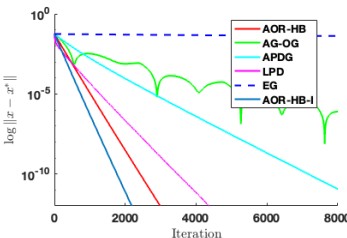

Figure 5: Simulation results for MSPBE (29).

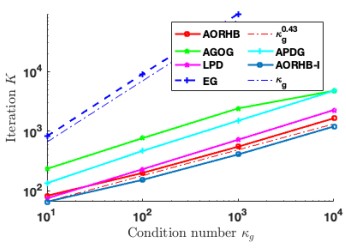

Figure 6: Rate of convergence.

We plot the convergence of error versus the number of iterations in Figure 5. Since each method has a comparable per-iteration computational cost, the iteration count serves as a fair measure of efficiency. We only plot the case $\kappa_g = 10^4$, but similar trends are observed for other condition numbers, where our AOR-HB-saddle methods consistently exhibit faster linear convergence compared to other algorithms.

Among the methods, AOR-HB-saddle-I, the semi-implicit scheme (26), achieves the fastest convergence and is overall the most time-efficient. It is preferable when the coupling term $\|B\|^2/(\mu_f\mu_g)$ exceeds $\sqrt{\kappa_f}$ and $\sqrt{\kappa_g}$, and the direct solver to compute $((1+\alpha)I + \frac{\alpha^2}{(1+\alpha)\mu_f\mu_g}BB^\top)^{-1}$ is efficient, which is the case ($n = 50$) here. When the direct solver becomes computationally expensive, the explicit scheme AOR-HB-saddle is preferable, provided an optimized step size $\alpha$ is used.

In Figure 6, we plot the iteration number $K$ versus $\kappa_g$ such that $\frac{\|x_K - x^*\|}{\|x_0 - x^*\|} \leq 10^{-6}$. In this log-log scale plot, we can observe that the growth of iteration complexity is $\mathcal{O}(\sqrt{\kappa_g})$ for accelerated algorithms and $\mathcal{O}(\kappa_g)$ for EG, which matches the convergence analysis.

Among all accelerated algorithms we have tested, our AOR-HB-saddle method requires fewer iteration steps to achieve the desired accuracy. In addition, AOR-HB-saddle has a single-loop structure and requires fewer parameters to tune, which is favorable for implementation.

## 5 CONCLUDING REMARKS

In conclusion, we have introduced the Accelerated Over-Relaxation Heavy-Ball (AOR-HB) methods, a significant advancement of the accelerated first order optimization algorithms. This fills a theoretical gap in heavy-ball momentum methods and opens the door to developing provable accelerated methods with potential forays into non-convex optimization scenarios.

AOR-HB comes with certain limitations. First, AOR-HB methods require the parameters $\mu$ and $L$. While this is typical for accelerated methods, exploring adaptive strategies to reduce parameter dependence is an interesting direction for future research. Second, the convergence rates established in some theorems are not as tight as those achieved by the TM method (Van Scoy et al., 2017; Taylor & Drori, 2023). Finally, investigating the algorithm's performance under stochastic conditions is essential for assessing its robustness in real-world machine learning applications.

ACKNOWLEDGEMENTS

The authors would like to thank Dr. Hao Luo and Professor Yurii Nesterov for their insightful discussions during the preparation of this work. We are also grateful to the anonymous reviewers for their constructive comments and suggestions, particularly for highlighting the Lifted Primal-Dual (LPD) method Thekumparampil et al. (2022), which helped improve the quality of the paper. We also thank Dr. Thekumparampil for generously providing the code for LPD. This work was supported by the National Science Foundation under grants DMS-2012465, DMS-2309777, and DMS-2309785.

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

## A  DEFINITIONS AND STANDARD RESULTS

We review some foundational results from the convex analysis.

Recall that the Bregman divergence of $f$ is defined as
$$D_f(y,x) := f(y) - f(x) - \langle \nabla f(x), y - x \rangle,$$
which is in general non-symmetric, i.e., $D_f(y,x) \neq D_f(x,y)$. A symmetrized Bregman divergence is defined as
$$\langle \nabla f(x) - \nabla f(y), x - y \rangle = D_f(y,x) + D_f(x,y).$$
We have the following bounds on the Bregman divergence and the symmetrized Bregman divergence.

**LEMMA A.1** (Section 2.1 in Nesterov (2018)). *Suppose* $f : \mathbb{R}^d \to \mathbb{R}$ *is* $\mu$*-strongly convex and* $L$*-smooth. For any* $x, y \in \mathbb{R}^d$,

$$\frac{\mu}{2}\|x - y\|^2 \leq D_f(y,x) \leq \frac{L}{2}\|x - y\|^2, \tag{30a}$$

$$\frac{1}{2L}\|\nabla f(x) - \nabla f(y)\|^2 \leq D_f(y,x) \leq \frac{1}{2\mu}\|\nabla f(x) - \nabla f(y)\|^2, \tag{30b}$$

$$\mu\|x - y\|^2 \leq \langle \nabla f(x) - \nabla f(y), x - y \rangle \leq L\|x - y\|^2, \tag{30c}$$

$$\frac{1}{L}\|\nabla f(x) - \nabla f(y)\|^2 \leq \langle \nabla f(x) - \nabla f(y), x - y \rangle \leq \frac{1}{\mu}\|\nabla f(x) - \nabla f(y)\|^2. \tag{30d}$$

The following three-point identity on the Bregman divergence will be used to replace the identity of squares.

**LEMMA A.2** (Bregman divergence identity (Chen & Teboulle, 1993)). *If function* $f : \mathbb{R}^d \to \mathbb{R}$ *is differentiable, then for any* $x, y, z \in \mathbb{R}^d$, *it holds that*
$$\langle \nabla f(y) - \nabla f(x), y - z \rangle = D_f(z,y) + D_f(y,x) - D_f(z,x). \tag{31}$$

*Proof.* By definition,
$$\begin{aligned} D_f(z,y) &= f(z) - f(y) - \langle \nabla f(y), z - y \rangle, \\ D_f(y,x) &= f(y) - f(x) - \langle \nabla f(x), y - x \rangle, \\ D_f(z,x) &= f(z) - f(x) - \langle \nabla f(x), z - x \rangle. \end{aligned}$$
Direct calculation gives the identity. □

## B RELATED WORKS

### B.1 EXTENSIONS/APPLICATIONS OF HEAVY-BALL METHODS

While the acceleration guarantee has not yet proved rigorously, applications of heavy-ball methods including extension to constrained and distributed optimization problems have confirmed its performance benefits over the standard gradient-based methods (Wang & Miller, 2013; Ochs et al., 2015; Ghadimi et al., 2013; Diakonikolas & Jordan, 2021).

Sutskever et al. (2013) showed that stochastic gradient descent with momentum improves the training of deep and recurrent neural networks. Also, using heavy-ball flow improves neural ODEs training and inference (Xia et al., 2021). Recently, accelerated convergence of stochastic heavy-ball methods has been established in Pan et al. (2023); Bollapragada et al. (2022); Wang et al. (2021) but only for quadratic objectives. To get rid of the hyperparameters used in the heavy-ball methods, Saab Jr et al. (2022) proposed an adaptive heavy-ball that estimates the Polyak's optimal hyper-parameters at each iteration.

### B.2 FIRST-ORDER METHODS AND DYNAMIC SYSTEM

One approach to better understand the mechanism of the iterative method is the continuous-time analysis: derive an ordinary differential equation (ODE) model which coincides with the iterative method taking step size close to zero. Starting from the accelerated first-order methods for unconstrained optimization, an important milestone in this direction is to understand the acceleration from the variational perspective (Su et al., 2016; Wibisono et al., 2016). While the iterative methods are first-order, the continuous-time dynamics proposed for accelerated methods are high-order or high-resolution ODEs (Shi et al., 2022; Sun et al., 2020; Muehlebach & Jordan, 2019; Attouch et al., 2000). With the continuous-time dynamic, novel accelerated methods are proposed by time discretization of the ODE model and usually the behaviour of the dynamic facilitates the convergence analysis of the iterative methods (Krichene et al., 2015; Luo & Chen, 2022; Aujol et al., 2022; Wilson et al., 2021; Siegel, 2019).

Due to the appealing results, systematic framework to draw connection between the dynamic system and the accelerated iterative method gains lots of interest (Scieur et al., 2017; Ushiyama et al., 2024; Taylor et al., 2018; Sanz Serna & Zygalakis, 2021). In fact, the theory and methods developed for other problem classes often build upon the work done in unconstrained optimization (Diakonikolas & Orecchia, 2019).

### B.3 EXTRA-GRADIENT METHODS BEYOND CONVEX OPTIMIZATION

The extra-gradient method, also known as optimistic gradient method can be formulated as:

$$x_{k+1} = x_k - \alpha_k(2F(x_k) - F(x_{k-1}))),$$

where $F(\cdot)$ is a (strongly) monotone and Liptschitz continuous operator. The origin of the extra-gradient method was proposed by (Popov, 1980; Korpelevich, 1976) for saddle point problems, in an equivalent form

$$\tilde{x}_k = x_k - F(\tilde{x}_{k-1}), \quad x_{k+1} = x_k - F(\tilde{x}_k).$$

The anchoring of the gradient was introduced to circumvent the issue that the standard gradient method does not always converge. This algorithm was subsequently studied by, to name just a few among others, Chambolle & Pock (2011); Malitsky (2015); Nesterov (2007) under various settings, and have recently attracted considerable interest for its wild application in machine learning such as training generative adversarial networks (Gidel et al., 2018; Daskalakis et al., 2017; Cui & Shanbhag, 2016; Peng et al., 2020). The optimal rate of convergence $\mathcal{O}(1/k)$ is achieved by (Korpelevich, 1976; Nemirovski, 2004; Gorbunov et al., 2022; Mokhtari et al., 2020b). When the operator is $\mu$-strongly monotone and $L$-Liptchitz continuous, the method was shown to exhibit a geometric convergence rate Tseng (1995); Nesterov & Scrimali (2006); Kotsalis et al. (2022); Song & Diakonikolas (2023). However, the iteration complexity of $\mathcal{O}(L/\mu|\log \epsilon|)$ is suboptimal when applied to convex optimization problems or saddle point problems.

### B.4 APPLICATIONS OF STRONGLY-CONVEX-STRONGLY CONCAVE SADDLE POINT PROBLEMS WITH BILINEAR COUPLING

A classical application is the regularized empirical risk minimization (ERM) with linear predictors, which is a classical supervised learning problem. Given a data matrix $B = [b_1, b_2, \cdots, b_n]^\top \in \mathbb{R}^{n \times m}$ where $b_i \in \mathbb{R}^m$ is the feature vector of the $i$-th data entry, the ERM problem aims to solve

$$\min_{u \in \mathbb{R}^m} g(Bu) + f(u), \tag{32}$$

where $g : \mathbb{R}^n \to \mathbb{R}$ is some strongly convex loss function, $f : \mathbb{R}^m \to \mathbb{R}$ is a strongly convex regularizer and $u \in \mathbb{R}^m$ is the linear predictor. Equivalently, we can solve the saddle point problem

$$\min_{u \in \mathbb{R}^m} \max_{p \in \mathbb{R}^n} \left\{ p^\top Bu - g^*(p) + f(u) \right\}. \tag{33}$$

The saddle point formulation is favorable in many scenarios, for instance (Zhang & Xiao, 2017; Du & Hu, 2019; Lei et al., 2017).

Another application is policy evaluation problems in reinforcement learning when finding minimum the mean squared projected Bellman error (Du et al., 2017):

$$\arg\min_u \frac{1}{2}\|Bu - b\|^2_{C^{-1}} + \frac{1}{2}\|u\|^2,$$

where $B, C$ are given matrices. The corresponding saddle point problem is

$$\min_{u \in \mathbb{R}^m} \max_{p \in \mathbb{R}^n} \frac{1}{2}\|u\|^2 - \frac{1}{2}\|p\|^2_C - \langle b, p \rangle + \langle Bu, p \rangle.$$

The saddle point formulation saves the computation cost of inverting $C$.

## C PROOF OF THEOREM 1.1

Consider the Lyapunov function

$$\mathcal{E}(\mathbf{z}) = \mathcal{E}(x, y) := f(x) - f(x^*) + \frac{\mu}{2}\|y - x^*\|^2 = D_f(x, x^*) + \frac{\mu}{2}\|y - x^*\|^2, \tag{34}$$

and the modified Lyapunov function

$$\begin{aligned} \mathcal{E}^\alpha(\mathbf{z}) = \mathcal{E}^\alpha(x, y) &:= \mathcal{E}(x, y) + \alpha\langle \nabla f(x) - \nabla f(x^*), y - x^* \rangle \\ &= D_f(x, x^*) + \frac{\mu}{2}\|y - x^*\|^2 + \alpha\langle \nabla f(x) - \nabla f(x^*), y - x^* \rangle. \end{aligned} \tag{35}$$

The novelty of $\mathcal{E}^\alpha(x, y)$ is the inclusion of the cross term $\alpha\langle \nabla f(x) - \nabla f(x^*), y - x^* \rangle$.

We denote the right-hand side of the accelerated gradient descent flow (6) as

$$\mathcal{G}(\mathbf{z}) = \mathcal{G}(x, y) := \begin{pmatrix} y - x, \\ x - y - \frac{1}{\mu}\nabla f(x) \end{pmatrix}.$$

**LEMMA C.1** (Strong Lyapunov property (Luo & Chen, 2022))**.**

$$-\langle \nabla\mathcal{E}(x, y), \mathcal{G}(x, y) \rangle \geq \mathcal{E}(x, y) + \frac{\mu}{2}\|x - y\|^2, \quad \forall\, x, y \in \mathbb{R}^d. \tag{36}$$

*Proof.* By direct calculation and the $\mu$-convexity of $f$,

$$\begin{aligned} -\langle \nabla\mathcal{E}(x, y), \mathcal{G}(x, y) \rangle &= \langle \nabla f(x), x - x^* \rangle - \mu\langle x - y, y - x^* \rangle \\ &= \langle \nabla f(x), x - x^* \rangle - \frac{\mu}{2}\left(\|x - x^*\|^2 - \|x - y\|^2 - \|y - x^*\|^2\right) \\ &\geq f(x) - f(x^*) + \frac{\mu}{2}\|y - x^*\|^2 + \frac{\mu}{2}\|x - y\|^2 = \mathcal{E}(x, y) + \frac{\mu}{2}\|x - y\|^2. \end{aligned}$$

$\square$

**LEMMA C.2.** *Suppose $f : \mathbb{R}^d \to \mathbb{R}$ is $\mu$-strongly convex and $L$-smooth. Denote by $\rho = \mu/L$. For any two vectors $\mathbf{z}_k, \mathbf{z}_{k+1} \in \mathbb{R}^{2d}$ and $\alpha > 0$, we have the following inequality on the Bregman divergence of $\mathcal{E}$ defined by (34)*

$$\left(1 - \frac{\alpha}{\sqrt{\rho}}\right) D_{\mathcal{E}}(\mathbf{z}_k, \mathbf{z}_{k+1}) \leq D_{\mathcal{E}}(\mathbf{z}_k, \mathbf{z}_{k+1}) \pm \alpha \langle y_{k+1} - y_k, \nabla f(x_{k+1}) - \nabla f(x_k) \rangle$$

$$\leq \left(1 + \frac{\alpha}{\sqrt{\rho}}\right) D_{\mathcal{E}}(\mathbf{z}_k, \mathbf{z}_{k+1}).$$

*Proof.* Using the identity of squares (for $y$) and Bregman divergence identity (31) (for $x$), we have the component form of

$$D_{\mathcal{E}}(\mathbf{z}_k, \mathbf{z}_{k+1}) = D_f(x_k, x_{k+1}) + \frac{\mu}{2} \|y_k - y_{k+1}\|^2.$$

By Cauchy-Schwarz inequality and bound (30b) in Lemma A.1, we have

$$\alpha \left| \langle y_{k+1} - y_k, \nabla f(x_{k+1}) - \nabla f(x_k) \rangle \right| \leq \frac{\alpha}{2\sqrt{\mu L}} \|\nabla f(x_k) - \nabla f(x_{k+1})\|^2 + \frac{\alpha\sqrt{L\mu}}{2} \|y_k - y_{k+1}\|^2$$

$$\leq \alpha \sqrt{\frac{L}{\mu}} D_f(x_k, x_{k+1}) + \frac{\alpha\sqrt{L\mu}}{2} \|y_k - y_{k+1}\|^2$$

$$= \alpha \sqrt{\frac{L}{\mu}} \left( D_f(x_k, x_{k+1}) + \frac{\mu}{2} \|y_k - y_{k+1}\|^2 \right).$$

$\square$

By Lemma C.2 and $\mathcal{E}(\mathbf{z}) = D_{\mathcal{E}}(\mathbf{z}, \mathbf{z}^*)$ where $\mathbf{z}^* := (x^*, x^*)^\top$, it is straightforward to verify the following bounds for $\mathcal{E}, \mathcal{E}^\alpha$ as Lyapunov functions. Assume $0 < \alpha \leq \sqrt{\rho}$ with $\rho = \frac{\mu}{L}$. Then

$$0 \leq \mathcal{E}^\alpha(\mathbf{z}) \leq 2\mathcal{E}(\mathbf{z}), \quad \forall \, \mathbf{z} = (x, y)^\top \in \mathbb{R}^{2d}. \tag{37}$$

Now we are in the position to prove our first main result.

*Proof of Theorem 1.1.* We write (14) as a correction of Implicit Euler discretization of (6):

$$\mathbf{z}_{k+1} - \mathbf{z}_k = \alpha \mathcal{G}(\mathbf{z}_{k+1}) - \alpha \begin{pmatrix} y_{k+1} - y_k \\ \frac{1}{\mu}(\nabla f(x_{k+1}) - \nabla f(x_k)) \end{pmatrix}. \tag{38}$$

and substitude into the identity of $\mathcal{E}$

$$\mathcal{E}(\mathbf{z}_{k+1}) - \mathcal{E}(\mathbf{z}_k) = \langle \nabla \mathcal{E}(\mathbf{z}_{k+1}), \mathbf{z}_{k+1} - \mathbf{z}_k \rangle - D_{\mathcal{E}}(\mathbf{z}_k, \mathbf{z}_{k+1})$$

$$= \alpha \langle \nabla \mathcal{E}(\mathbf{z}_{k+1}), \mathcal{G}(\mathbf{z}_{k+1}) \rangle - \alpha \langle \nabla \mathcal{E}(\mathbf{z}_{k+1}), \begin{pmatrix} y_{k+1} - y_k \\ \frac{1}{\mu}(\nabla f(x_{k+1}) - \nabla f(x_k)) \end{pmatrix} \rangle - D_{\mathcal{E}}(\mathbf{z}_k, \mathbf{z}_{k+1}).$$

We write out the component form of the middle term and split it as

$$\alpha \langle \nabla f(x_{k+1}) - \nabla f(x^*), y_{k+1} - y_k \rangle + \alpha \langle y_{k+1} - x^*, \nabla f(x_{k+1}) - \nabla f(x_k) \rangle$$

$$= \alpha \langle \nabla f(x_{k+1}) - \nabla f(x^*), y_{k+1} - x^* \rangle + \alpha \langle y_{k+1} - y_k, \nabla f(x_{k+1}) - \nabla f(x_k) \rangle$$

$$- \alpha \langle y_k - x^*, \nabla f(x_k) - \nabla f(x^*) \rangle.$$

Substitute back to (18) and rearrange terms, we obtain the following identity:

$$\mathcal{E}^\alpha(\mathbf{z}_{k+1}) - \mathcal{E}^\alpha(\mathbf{z}_k) = \alpha \langle \nabla \mathcal{E}(\mathbf{z}_{k+1}), \mathcal{G}(\mathbf{z}_{k+1}) \rangle$$

$$- D_{\mathcal{E}}(\mathbf{z}_k, \mathbf{z}_{k+1}) - \alpha \langle y_{k+1} - y_k, \nabla f(x_{k+1}) - \nabla f(x_k) \rangle, \tag{39}$$

By Lemma C.2, we can drop the terms in the second line for $0 \leq \alpha \leq \sqrt{\rho}$. Use the strong Lyapunov property (36) and bound (37), we have

$$\mathcal{E}^\alpha(\mathbf{z}_{k+1}) - \mathcal{E}^\alpha(\mathbf{z}_k) \leq -\alpha \mathcal{E}(\mathbf{z}_{k+1}) \leq -\frac{\alpha}{2} \mathcal{E}^\alpha(\mathbf{z}_{k+1}), \tag{40}$$

which implies the global linear convergence of $\mathcal{E}^\alpha$:

$$\mathcal{E}^\alpha(\mathbf{z}_{k+1}) \leq \frac{1}{1+\alpha/2}\mathcal{E}^\alpha(\mathbf{z}_k) \leq \left(\frac{1}{1+\alpha/2}\right)^{k+1}\mathcal{E}^\alpha(\mathbf{z}_0), \quad k \geq 0.$$

Moreover, (40) implies

$$\mathcal{E}(\mathbf{z}_{k+1}) \leq \frac{1}{\alpha}(\mathcal{E}^\alpha(\mathbf{z}_k) - \mathcal{E}^\alpha(\mathbf{z}_{k+1})) \leq \frac{1}{\alpha}\mathcal{E}^\alpha(\mathbf{z}_k) \leq \frac{1}{\alpha}\left(\frac{1}{1+\alpha/2}\right)^k \mathcal{E}^\alpha(\mathbf{z}_0),$$

which coincides with (8) with $C_0 = \frac{1}{\alpha}\mathcal{E}^\alpha(\mathbf{z}_0) \geq 0$. $\qquad\square$

## D  AOR-HB METHOD FOR NON-STRONGLY CONVEX MINIMIZATION

In this section, we present AOR-HB for non-strongly convex case, i.e., $\mu = 0$. The convergence analysis is motivated by the Hessian-driven Nesterov accelerated gradient (HNAG) flow proposed in Chen & Luo (2021).

### D.1  AOR-HB-0 METHOD

Consider the rescaled rotated gradient flow

$$x' = y - x,$$
$$y' = -\frac{1}{\gamma}\nabla f(x),$$

where $\gamma > 0$ is a time rescaling parameter. This is the degenerated version of the rotated gradient flow Chen & Luo (2021) for non-strongly convex cases.

Using AOR technique to discretize, we get the update rule: given $(x_k, y_k)$, compute

$$\begin{aligned}\frac{x_{k+1} - x_k}{\alpha_k} &= y_k - x_{k+1},\\ \frac{y_{k+1} - y_k}{\alpha_k} &= -\frac{1}{\gamma_k}(2\nabla f(x_{k+1}) - \nabla f(x_k)).\end{aligned} \tag{41}$$

We shall choose the parameters

$$\alpha_k = \frac{2}{k+1}, \quad \gamma_k = \alpha_k^2 L.$$

By eliminating $y_k$'s, we get the AOR-HB-0 method:

$$x_{k+1} = x_k - \frac{k}{k+3}\frac{1}{L}\left(2\nabla f(x_k) - \nabla f(x_{k-1})\right) + \frac{k}{k+3}(x_k - x_{k-1}).$$

### D.2  EQUIVALENCE TO HNAG

Introduce another parameter $\beta_k > 0$ and change of variable

$$y_k = v_k - \beta_k \nabla f(x_k). \tag{42}$$

Substitute back $y_k$ to (41), we obtain discretization in terms of $(x_k, v_k)$:

$$\begin{aligned}\frac{x_{k+1} - x_k}{\alpha_k} &= v_k - x_{k+1} - \beta_k \nabla f(x_k),\\ \frac{v_{k+1} - v_k}{\alpha_k} &= -\frac{1}{\gamma_k}\left(2 - \frac{\gamma_k \beta_{k+1}}{\alpha_k}\right)\nabla f(x_{k+1}) + \left(\frac{1}{\gamma_k} - \frac{\beta_k}{\alpha_k}\right)\nabla f(x_k).\end{aligned} \tag{43}$$

Now we choose $\beta_k = \frac{\alpha_k}{\gamma_k}$ such that $\alpha_k \beta_k = \frac{\alpha_k^2}{\gamma_k} = \frac{1}{L}$. We can further simplify (43) to

$$\frac{x_{k+1} - x_k}{\alpha_k} = v_k - x_{k+1} - \beta_k \nabla f(x_k), \tag{44a}$$

$$\frac{v_{k+1} - v_k}{\alpha_k} = -\frac{1}{\tilde{\gamma}_k}\nabla f(x_{k+1}), \tag{44b}$$

with

$$\tilde{\gamma}_k = \left(2 - \frac{\gamma_k \beta_{k+1}}{\alpha_k}\right)^{-1} \gamma_k = \left(2 - \frac{\alpha_k}{\alpha_{k+1}}\right)^{-1} \gamma_k = \frac{k+1}{k} \gamma_k.$$

It is straight forward to verity that $\tilde{\gamma}_k$ satisfies

$$\tilde{\gamma}_{k+1} = \frac{k+2}{k+1} \gamma_{k+1} = \frac{k+2}{k+1} \frac{\alpha_{k+1}^2}{\alpha_k^2} \gamma_k = \frac{k+1}{k+2} \gamma_k \leq \frac{(k+1)^2}{k(k+3)} \gamma_k = \frac{1}{1+\alpha_k} \tilde{\gamma}_k,$$

which is equivalent to

$$\frac{\tilde{\gamma}_{k+1} - \tilde{\gamma}_k}{\alpha_k} \leq -\tilde{\gamma}_{k+1}. \tag{45}$$

Therefore, (44) combining (45) is a discretization of the following variant of the HNAG flow:

$$\begin{aligned}
x' &= v - x - \beta \nabla f(x), \\
v' &= -\frac{1}{\tilde{\gamma}} \nabla f(x), \\
\tilde{\gamma}' &\leq -\tilde{\gamma}.
\end{aligned} \tag{46}$$

Notice that the flow for the dynamic coefficient is $\gamma' = -\gamma$ in Chen & Luo (2021). We shall show the convergence rate is accelerated for the scheme formulated in (44). The proofs turned out to be similar.

### D.3 CONVERGENCE ANALYSIS

Define the Lyapunov function

$$\mathcal{E}(x, v, \tilde{\gamma}) := f(x) - f(x^*) + \frac{\tilde{\gamma}}{2} \|v - x^*\|^2.$$

We denote the right hand side of the flow (46) as $\mathcal{G}(x, v, \tilde{\gamma})$. Direct computation gives

$$\begin{aligned}
-\langle \nabla \mathcal{E}(x, v, \tilde{\gamma}), \mathcal{G}(x, v, \tilde{\gamma}) \rangle &= \langle \nabla f(x), x - x^* \rangle + \beta \|\nabla f(x)\|^2 + \frac{\tilde{\gamma}}{2} \|v - x^*\|^2 \\
&\geqslant \mathcal{E}(x, v, \tilde{\gamma}) + \beta \|\nabla f(x)\|^2.
\end{aligned}$$

Hence $\mathcal{E}(\cdot)$ satisfies strong Lyapunov property.

**THEOREM D.1** (Convergence rate for non-strongly convex minimization). *For $(x_k, v_k)$ generated by (44) with initial values $(x_1, v_1, \tilde{\gamma}_1) = (x_0, v_0, \tilde{\gamma}_0)$ and*

$$\alpha_k = \frac{2}{k+1}, \quad \beta_k = \frac{1}{\alpha_k L}, \quad \tilde{\gamma}_k = \frac{k+1}{k} \alpha_k^2 L, \quad k \geq 1$$

*we have the convergence*

$$\mathcal{E}(x_{k+1}, v_{k+1}, \tilde{\gamma}_{k+1}) \leq \prod_{j=1}^{k} \frac{1}{1+\alpha_j} \mathcal{E}_0 \leq \frac{6}{(k+3)(k+2)} \mathcal{E}_0, \tag{47}$$

*where $\mathcal{E}_0 = \mathcal{E}(x_0, v_0, \tilde{\gamma}_0)$.*

*Proof.* For short, we denote $\mathbf{z}_k = (x_k, v_k, \tilde{\gamma}_k)$ and the right hand side of (46) by $\mathcal{G}(\mathbf{z}_k) := [\mathcal{G}_k^x, \mathcal{G}_k^v, \mathcal{G}_k^\gamma]^\top$. Let us calculate the difference $\mathcal{E}(\mathbf{z}_{k+1}) - \mathcal{E}(\mathbf{z}_k) = I_1 + I_2 + I_3$ with

$$\begin{cases}
I_1 := \mathcal{E}(x_{k+1}, v_k, \tilde{\gamma}_k) - \mathcal{E}(\mathbf{z}_k), \\
I_2 := \mathcal{E}(x_{k+1}, v_{k+1}, \tilde{\gamma}_k) - \mathcal{E}(x_{k+1}, v_k, \tilde{\gamma}_k), \\
I_3 := \mathcal{E}(\mathbf{z}_{k+1}) - \mathcal{E}(x_{k+1}, v_{k+1}, \tilde{\gamma}_k).
\end{cases}$$

We shall estimate the above three terms one by one.

As $\mathcal{E}$ is linear in terms of $\tilde{\gamma}$, we get

$$\mathrm{I}_3 = \langle \nabla_\gamma \mathcal{E}\left(x_{k+1}, v_{k+1}, \tilde{\gamma}_{k+1}\right), \gamma_{k+1} - \gamma_k \rangle = \alpha_k \langle \nabla_\gamma \mathcal{E}\left(x_{k+1}, v_{k+1}, \tilde{\gamma}_{k+1}\right), \mathcal{G}_{k+1}^\gamma \rangle.$$

For the second item $\mathrm{I}_2$, we use the fact that $\mathcal{E}\left(x_{k+1}, \cdot, \tilde{\gamma}_k\right)$ is quadratic and equation (44b) to get

$$\mathrm{I}_2 = \langle \nabla_v \mathcal{E}\left(x_{k+1}, v_{k+1}, \tilde{\gamma}_k\right), v_{k+1} - v_k \rangle - \frac{\tilde{\gamma}_k}{2}\left\| v_{k+1} - v_k \right\|^2$$

$$= \alpha_k \langle \nabla_v \mathcal{E}\left(x_{k+1}, v_{k+1}, \tilde{\gamma}_{k+1}\right), \mathcal{G}_{k+1}^v \rangle - \frac{\tilde{\gamma}_k}{2}\left\| v_{k+1} - v_k \right\|^2,$$

where in the last step, we switch the variable $(x_{k+1}, v_{k+1}\tilde{\gamma}_k)$ to $(x_{k+1}, v_{k+1}, \tilde{\gamma}_{k+1})$ as the parameter $\tilde{\gamma}$ is canceled in the product $\langle \nabla_v \mathcal{E}, \mathcal{G}^v \rangle$.

Next for $\mathrm{I}_1$ we use the update formula (44a) and bound (30b),

$$\begin{aligned}
\mathrm{I}_1 = &\ \langle \nabla_x \mathcal{E}\left(x_{k+1}, v_k, \tilde{\gamma}_{k+1}\right), x_{k+1} - x_k \rangle - D_f(x_k, x_{k+1}) \\
\le &\ \alpha_k \langle \nabla_x \mathcal{E}\left(\mathbf{z}_{k+1}\right), \mathcal{G}^x\left(\mathbf{z}_{k+1}\right) \rangle + \alpha_k \beta_k \langle \nabla f\left(x_{k+1}\right), \nabla f\left(x_{k+1}\right) - \nabla f\left(x_k\right) \rangle \\
&\ + \alpha_k \langle \nabla f\left(x_{k+1}\right), v_k - v_{k+1} \rangle - \frac{1}{2L}\left\| \nabla f\left(x_{k+1}\right) - \nabla f\left(x_k\right) \right\|^2.
\end{aligned} \tag{48}$$

In the first term, we can switch $(x_{k+1}, v_k, \tilde{\gamma}_k)$ to $\mathbf{z}_{k+1}$ because $\nabla_x \mathcal{E} = \nabla f(x)$ is independent of $(v, \gamma)$.

We use Cauchy-Schwarz inequality to bound the terms in (48) as follows:

$$\alpha_k \left\| \nabla f\left(x_{k+1}\right) \right\| \left\| v_k - v_{k+1} \right\| \le \frac{1}{2L}\left\| \nabla f\left(x_{k+1}\right) \right\|^2 + \frac{\alpha_k^2 L}{2}\left\| v_k - v_{k+1} \right\|^2,$$

and

$$\begin{aligned}
&\alpha_k \beta_k \langle \nabla f\left(x_{k+1}\right), \nabla f\left(x_{k+1}\right) - \nabla f\left(x_k\right) \rangle \\
&= -\frac{\alpha_k \beta_k}{2}\left\| \nabla f\left(x_k\right) \right\|^2 + \frac{\alpha_k \beta_k}{2}\left\| \nabla f\left(x_{k+1}\right) \right\|^2 + \frac{\alpha_k \beta_k}{2}\left\| \nabla f\left(x_{k+1}\right) - \nabla f\left(x_k\right) \right\|^2.
\end{aligned}$$

Adding all together and applying strong Lyapunov property at $\mathbf{z}_{k+1}$ (but with $\beta_k$ not $\beta_{k+1}$) yields that

$$\begin{aligned}
\mathcal{E}(\mathbf{z}_{k+1}) - \mathcal{E}(\mathbf{z}_k) \le &\ -\alpha_k \mathcal{E}(\mathbf{z}_{k+1}) - \frac{1}{2}\left(\tilde{\gamma}_k - \alpha_k^2 L\right)\left\| v_k - v_{k+1} \right\|^2 - \frac{\alpha_k \beta_k}{2}\left\| \nabla f\left(x_k\right) \right\|^2 \\
&\ + \frac{1}{2}\left(\frac{1}{L} - \alpha_k \beta_k\right)\left\| \nabla f\left(x_{k+1}\right) \right\|^2 \\
&\ + \frac{1}{2}\left(\alpha_k \beta_k - \frac{1}{L}\right)\left\| \nabla f\left(x_{k+1}\right) - \nabla f\left(x_k\right) \right\|^2
\end{aligned}$$

By our choice of parameters:

$$\alpha_k \beta_k - \frac{1}{L} = 0, \quad \tilde{\gamma}_k > \gamma = \alpha_k^2 L,$$

we get

$$\mathcal{E}(\mathbf{z}_{k+1}) - \mathcal{E}(\mathbf{z}_k) \le -\alpha_k \mathcal{E}(\mathbf{z}_{k+1})$$

by throwing away negative terms. Rearrange terms and by induction we have the decay (47).

The convergence rate

$$\prod_{j=1}^k \frac{1}{1 + \alpha_j} = \prod_{j=1}^k \frac{j+1}{j+3} = \frac{6}{(k+3)(k+2)} = \mathcal{O}\left(\frac{1}{k^2}\right).$$

We conclude that for AOR-HB-0 method (9), $f(x) - f(x^*)$ converges with complexity $\mathcal{O}(1/\sqrt{\epsilon})$.

$\square$

# E    STRONGLY LYAPUNOV PROPERTY FOR AOR-HB-COMPOSITE FLOW

Consider the Lyapunov function

$$\mathcal{E}(x,y) := D_f(x,x^*) + \frac{\mu}{2}\|y - x^*\|^2,$$

where the Bregman divergence $D_f(x,x^*)$ is generalization of $f(x) - f(x^*)$ in the single convex function case.

We rewrite the flow (23) as

$$x' = y - x,$$
$$y' = x - y - \frac{1}{\mu}(\nabla f(x) + \xi), \quad \xi \in \partial g(y).$$

With this formulation, we denote the right-hand side as

$$\mathcal{G}(\mathbf{z}) = \mathcal{G}(x,y) := \begin{pmatrix} y - x, \\ x - y - \frac{1}{\mu}\nabla(f(x) + \xi) \end{pmatrix}.$$

Using the first order optimality condition, $\mathcal{G}(\mathbf{z}^*) = 0$ for some $\xi^* \in \partial g(x^*)$.

We restate and prove the strong Lyapunov property in the following Lemma. The difference compared with the proof of Lemma C.1 is highlighted in blue.

**LEMMA E.1** (Strong Lyapunov property for AOR-HB-composite flow (23)).

$$-\langle \nabla\mathcal{E}(x,y), \mathcal{G}(x,y)\rangle \geq \mathcal{E}(x,y) + \frac{\mu}{2}\|x - y\|^2, \quad \forall\, x, y \in \mathbb{R}^d.$$

*Proof.* By direct calculation and the $\mu$-convexity of $f$,

$$-\langle \nabla\mathcal{E}(x,y), \mathcal{G}(x,y)\rangle = -\langle \nabla\mathcal{E}(x,y), \mathcal{G}(x,y) - \mathcal{G}(x^*,x^*)\rangle$$
$$= \langle \nabla f(x) - \nabla f(x^*), x - x^*\rangle - \mu\langle x - y, y - x^*\rangle + \langle \xi - \xi^*, y - x^*\rangle$$
$$= \langle \nabla f(x) - \nabla f(x^*), x - x^*\rangle - \frac{\mu}{2}\left(\|x - x^*\|^2 - \|x - y\|^2 - \|y - x^*\|^2\right) + \langle \xi - \xi^*, y - x^*\rangle$$
$$\geq D_f(x,x^*) + \frac{\mu}{2}\|y - x^*\|^2 + \frac{\mu}{2}\|x - y\|^2 = \mathcal{E}(x,y) + \frac{\mu}{2}\|x - y\|^2,$$

where the inequality used the convexity of $g$:

$$\langle \xi - \eta, y - x\rangle \geq 0, \quad \forall\, x, y \in \mathbb{R}^d \text{ and } \xi \in \partial g(y), \eta \in \partial g(x).$$

$\square$

**THEOREM E.1** (Convergence rate of AOR-HB method for composite convex minimization). *Suppose $f$ is $\mu$-strongly convex and $L$-smooth and $g$ is non-smooth and convex. Let $(x_k, y_k)$ be generated by Algorithm 2 with initial value $(x_0, y_0)$ and step size $\alpha = \sqrt{\mu/L}$. Then there exists a non-negative constant $C_0 = C_0(x_0, y_0, \mu, L)$ so that we have the accelerated linear convergence*

$$D_f(x,x^*) + \frac{\mu}{2}\|y_{k+1} - x^*\|^2 \leq C_0\left(\frac{1}{1 + \frac{1}{2}\sqrt{\mu/L}}\right)^k, \quad k \geq 1. \tag{49}$$

*Proof.* See Appendix C.

$\square$

## F    EQUIVALENT FORMULATION OF AOR-HB-SADDLE

Algorithm 2 is equivalent to the following discretization of the HB-saddle flow (56):

$$\frac{u_{k+1} - u_k}{\alpha} = v_k - u_{k+1}, \tag{50a}$$

$$\frac{p_{k+1} - p_k}{\alpha} = q_k - p_{k+1}, \tag{50b}$$

$$\frac{v_{k+1} - v_k}{\alpha} = u_{k+1} - v_{k+1} - \frac{1}{\mu_f}\left(2\nabla f(u_{k+1}) - \nabla f(u_k) + B^\top q_k\right), \tag{50c}$$

$$\frac{q_{k+1} - q_k}{\alpha} = p_{k+1} - q_{k+1} - \frac{1}{\mu_g}\left(2\nabla g(p_{k+1}) - \nabla g(p_k) - B(2v_{k+1} - v_k)\right). \tag{50d}$$

The discretization is a mixture of implicit Euler and explicit Euler with time step size $\alpha$. For both the gradient terms and $Bv$ term, we use the AOR technique, i.e., $\nabla f(u) \approx 2\nabla f(u_{k+1}) - \nabla f(u_k)$, $\nabla g(p) \approx 2\nabla g(p_{k+1}) - \nabla g(p_k)$ and $Bv \approx B(2v_{k+1} - v_k)$. Algorithm 3 is implementation-friendly while (50) is convenient for deriving convergence analysis.

## G    PROOFS OF SECTION 3

### G.1    A CLASS OF MONOTONE OPERATOR EQUATIONS

In fact, we can extend HB flow to a broad class of monotone operator equation $\mathcal{A}(x) = 0$ with

$$\mathcal{A}(x) = \nabla F(x) + \mathcal{N}x, \tag{51}$$

where $F$ is a strongly convex function and $L_F$ smooth, and $\mathcal{N}$ is a linear and skew-symmetric operator, i.e., $\mathcal{N}^\top = -\mathcal{N}$. Then $\mathcal{A}$ is Lipschitz continuous with constant $L_\mathcal{A} \leq L_F + \|\mathcal{N}\|$. Therefore $\mathcal{A}$ is monotone and Lipschitz continuous which is also known as inverse-strongly monotonicity (Browder & Petryshyn, 1967; Liu & Nashed, 1998) or co-coercivity (Zhu & Marcotte, 1996). Consequently equation $\mathcal{A}(x) = 0$ has a unique solution $x^*$ (Rockafellar, 1976).

As a special case, we recover strongly-convex-strongly-concave saddle point problems with bilinear coupling when $x := (u, p)^\top$, $F(x) := f(u) + g(p)$ and $\mathcal{N} := \begin{pmatrix} 0 & B^\top \\ -B & 0 \end{pmatrix}$.

We introduce an accelerated gradient flow

$$\begin{cases} x' = y - x, \\ y' = x - y - \mu^{-1}(\nabla F(x) + \mathcal{N}y). \end{cases} \tag{52}$$

Comparing with the accelerated gradient flow (6) for convex optimization, the difference is the gradient and skew-symmetric splitting $\mathcal{A}(x) \to \nabla F(x) + \mathcal{N}y$.

Denote the vector field on the right hand side of (52) by $\mathcal{G}(x, y)$. Then $\mathcal{G}(x^*, x^*) = 0$ and thus $(x^*, x^*)$ is an equilibrium point of (52).

We first show $(x^*, x^*)$ is exponentially stable. Consider the Lyapunov function:

$$\mathcal{E}(x, y) = D_F(x, x^*) + \frac{\mu}{2}\|y - x^*\|^2. \tag{53}$$

For $\mu$-strongly convex $F$, function $D_F(\cdot, x^*) \in \mathcal{S}_\mu$. Then $\mathcal{E}(x, y) \geq 0$ and $\mathcal{E}(x, y) = 0$ iff $x = y = x^*$.

We then verify the strong Lyapunov property. The proof is similar to that of Lemma C.1.

**THEOREM G.1** (Strong Lyapunov Property). *Assume function $F$ is $\mu$-strongly convex. Then for the Lyapunov function (53) and the accelerated gradient flow vector field $\mathcal{G}$ asoociated to (52), the following strong Lyapunov property holds*

$$-\nabla \mathcal{E}(x, y) \cdot \mathcal{G}(x, y) \geq \mathcal{E}(x, y) + \frac{\mu}{2}\|y - x\|^2, \quad \forall x, y \in \mathcal{V}. \tag{54}$$

*Proof.* First of all, as $\mathcal{G}(x^*, x^*) = 0$,
$$-\langle \nabla \mathcal{E}(x, y), \mathcal{G}(x, y) \rangle = -\langle \nabla \mathcal{E}(x, y), \mathcal{G}(x, y) - \mathcal{G}(x^*, x^*) \rangle.$$
Direct computation gives
$$-\langle \nabla \mathcal{E}(x, y), \mathcal{G}(x, y) \rangle = \langle \nabla D_F(x, x^*), x - x^* - (y - x^*) \rangle - \mu \langle y - x^*, x - x^* \rangle$$
$$+\mu \|y - x^*\|^2 + \langle \nabla F(x) - \nabla F(x^*), y - x^* \rangle + \langle y - x^*, \mathcal{N}(y - x^*) \rangle$$
$$= \langle \nabla F(x) - \nabla F(x^*), x - x^* \rangle + \mu \|y - x^*\|^2 - \mu \langle y - x^*, x - x^* \rangle,$$
where we have used $\nabla D_F(x, x^*) = \nabla F(x) - \nabla F(x^*)$ and $\langle y - x^*, \mathcal{N}(y - x^*) \rangle = 0$ since $\mathcal{N}$ is skew-symmetric. We expand the last two term using the identity of squares:
$$\frac{1}{2} \|y - x^*\|^2 - \langle y - x^*, x - x^* \rangle = \frac{1}{2} \|y - x\|^2 - \frac{1}{2} \|x - x^*\|^2.$$
Using the bound (30a), we get
$$\langle \nabla F(x) - \nabla F(x^*), x - x^* \rangle = D_F(x, x^*) + D_F(x^*, x) \geq D_F(x, x^*) + \frac{\mu}{2} \|x - x^*\|^2$$
and obtain (54). $\qquad \square$

The calculation is more clear when $\nabla F(x) = Ax$ is linear with $A = \nabla^2 F \geq \mu I$. We denote by $\mathbf{z} = (x, y)^\top$ and $\mathcal{E}(\mathbf{z}) = \frac{1}{2} \|\mathbf{z} - \mathbf{z}^*\|_{\mathcal{D}}^2$ with $\mathcal{D} = \begin{pmatrix} A & 0 \\ 0 & \mu I \end{pmatrix}$. Then $-\langle \nabla \mathcal{E}(x, y), \mathcal{G}(x, y) - \mathcal{G}(x^*, x^*) \rangle$ is a quadratic form $(\mathbf{z} - \mathbf{z}^*)^\top \mathcal{M}(\mathbf{z} - \mathbf{z}^*)$. We calculate the matrix $\mathcal{M}$ as
$$\begin{pmatrix} A & 0 \\ 0 & \mu I \end{pmatrix} \begin{pmatrix} I & -I \\ -I + \mu^{-1} A & I + \mu^{-1} \mathcal{N} \end{pmatrix} = \begin{pmatrix} A & -A \\ -\mu I + A & \mu I + \mathcal{N} \end{pmatrix}.$$
As $v^\top M v = v^\top \mathrm{sym}(M) v$ where $\mathrm{sym}(\cdot)$ is the symmetric part of a matrix, direct computation gives
$$\mathrm{sym} \begin{pmatrix} A & -A \\ -\mu I + A & \mu I + \mathcal{N} \end{pmatrix} = \begin{pmatrix} A & -\mu I/2 \\ -\mu I/2 & \mu I \end{pmatrix}$$
$$\geq \begin{pmatrix} A/2 & 0 \\ 0 & \mu I/2 \end{pmatrix} + \frac{1}{2} \begin{pmatrix} \mu I & -\mu I \\ -\mu I & \mu I \end{pmatrix},$$
where in the last step we use the convexity $A \geq \mu I$. Then (54) follows.

## G.2 EXPONENTIAL STABILITY OF HB-SADDLE FLOW

Return to the saddle point problems, we let $\mathbf{z} = (x, y)^\top$ with $x = (u, p)^\top, y = (v, q)^\top \in \mathbb{R}^m \times \mathbb{R}^n$ and denote by
$$F(x) := f(u) + g(p), \quad \mathcal{N} := \begin{pmatrix} 0 & B^\top \\ -B & 0 \end{pmatrix}, \quad \mathcal{D}_\mu := \begin{pmatrix} \mu_f & 0 \\ 0 & \mu_g \end{pmatrix}.$$

Consider the Lyapunov function
$$\mathcal{E}(\mathbf{z}) = \mathcal{E}(u, v, p, q) := D_F(x, x^*) + \frac{1}{2} \|y - x^*\|_{\mathcal{D}_\mu}^2$$
$$= D_f(u, u^*) + D_g(p, p^*) + \frac{\mu_f}{2} \|v - u^*\|^2 + \frac{\mu_g}{2} \|q - p^*\|^2, \tag{55}$$

As $f, g$ are strongly convex, $\mathcal{E}(\mathbf{z}) \geq 0$ and $\mathcal{E}(\mathbf{z}) = 0$ iff $u = v = u^*$ and $p = q = p^*$. Recall that the HB-saddle flow is
$$u' = v - u,$$
$$p' = q - p,$$
$$v' = u - v - \frac{1}{\mu_f} (\nabla f(u) + B^\top q), \tag{56}$$
$$q' = p - q - \frac{1}{\mu_g} (\nabla g(p) - Bv).$$
Denoted the vector field on the right hand side of (56) by $\mathcal{G}(u, v, p, q)$, as a special case of Theorem G.1, we obtain the following strong Lyapunov property.

**THEOREM G.2** (Strong Lyapunov property for HB-saddle flow). *Suppose $f$ is $\mu_f$-strongly convex $g$ is $\mu_g$-strongly convex. The following strong Lyapunov property holds for all $u, v \in \mathbb{R}^m, p, q \in \mathbb{R}^n$.*

$$-\langle \nabla \mathcal{E}(u, v, p, q), \mathcal{G}(u, v, p, q) \rangle \geq \mathcal{E}(u, v, p, q) + \frac{\mu_f}{2}\|v - u\|^2 + \frac{\mu_g}{2}\|q - p\|^2. \qquad (57)$$

*Consequently, for solutions $(u(t), v(t), p(t), q(t))$ of the HB-saddle flow (25), we have the exponential stability:*

$$D_f(u, u^*) + D_g(p, p^*) + \frac{\mu_f}{2}\|v - u^*\|^2 + \frac{\mu_g}{2}\|q - p^*\|^2 \leq e^{-t}\mathcal{E}(u(0), v(0), p(0), q(0)), \quad t > 0.$$

### G.3 PROOF OF THEOREM 1.2

Consider the modified Lyapunov function

$$\mathcal{E}^\alpha(\mathbf{z}) := \mathcal{E}(\mathbf{z}) + \alpha\langle \nabla F(x) - \nabla F(x^*), y - x^* \rangle - \alpha\|y - y^*\|^2_{\mathcal{B}^{\mathrm{sym}}}, \qquad (58)$$

where $\mathcal{E}(\mathbf{z})$ is defined as (55) and $\mathcal{B}^{\mathrm{sym}} = \begin{pmatrix} 0 & B^\top \\ B & 0 \end{pmatrix}$. Due to the bilinear coupling, additional cross term $\alpha\|y - y^*\|^2_{\mathcal{B}^{\mathrm{sym}}} = 2\alpha(B(v - v^*), q - q^*)$ is included in $\mathcal{E}^\alpha$.

We shall split $\frac{1}{2}\|y - x^*\|^2_{\mathcal{D}_\mu}$ to $\frac{\beta}{2}\|y - x^*\|^2_{\mathcal{D}_\mu} + \frac{1-\beta}{2}\|y - x^*\|^2_{\mathcal{D}_\mu}$ to bound the two cross terms. The cross term $\alpha\langle \nabla F(x) - \nabla F(x^*), y - x^* \rangle$ can be bounded using the vector form of Lemma C.2. Next we focus on the $\alpha\|y - y^*\|^2_{\mathcal{B}^{\mathrm{sym}}}$.

**LEMMA G.1.** *Denoted by $\mathcal{D}_\mu = \begin{pmatrix} \mu_f I & 0 \\ 0 & \mu_g I \end{pmatrix}$ with $\mu_f, \mu_g > 0$ and $\mathcal{B}^{\mathrm{sym}} = \begin{pmatrix} 0 & B^\top \\ B & 0 \end{pmatrix}$. For any $\alpha, c > 0$, we have*

$$\left(1 - \frac{\alpha\|B\|}{c\sqrt{\mu_f\mu_g}}\right) c\mathcal{D}_\mu \leq c\mathcal{D}_\mu \pm \alpha\mathcal{B}^{\mathrm{sym}} \leq \left(1 + \frac{\alpha\|B\|}{c\sqrt{\mu_f\mu_g}}\right) c\mathcal{D}_\mu. \qquad (59)$$

*Proof.* We first calculate the eigenvalues of $\mathcal{D}_\mu^{-1}\mathcal{B}^{\mathrm{sym}}$. By choosing the SVD basis of $B = U\Sigma V$, its eigenvalue is given by the $2 \times 2$ matrix $\begin{pmatrix} 0 & \mu_f^{-1}\sigma(B) \\ \mu_g^{-1}\sigma(B) & 0 \end{pmatrix}$, where $\sigma(B)$ is a singular value of $B$. So $\lambda(\mathcal{D}_\mu^{-1}\mathcal{B}^{\mathrm{sym}}) = \pm\frac{\sigma(B)}{\sqrt{\mu_f\mu_g}}$ and consequently

$$-\frac{\|B\|}{\sqrt{\mu_f\mu_g}} \leq \lambda_{\min}(\mathcal{D}_\mu^{-1}\mathcal{B}^{\mathrm{sym}}) \leq \lambda_{\max}(\mathcal{D}_\mu^{-1}\mathcal{B}^{\mathrm{sym}}) \leq \frac{\|B\|}{\sqrt{\mu_f\mu_g}}. \qquad (60)$$

Then write $c\mathcal{D}_\mu \pm \alpha\mathcal{B}^{\mathrm{sym}} = c\mathcal{D}_\mu(I \pm \frac{\alpha}{c}\mathcal{D}_\mu^{-1}\mathcal{B}^{\mathrm{sym}})$ and apply the bound (60) to get the desired result. $\qquad \square$

**LEMMA G.2.** *Suppose $f : \mathbb{R}^m \to \mathbb{R}$ is $\mu_f$-strongly convex and $L_f$-smooth, $g : \mathbb{R}^n \to \mathbb{R}$ is $\mu_g$-strongly convex and $L_g$-smooth. Let $F(\mathbf{z}) = f(u) + g(p)$. For any $\beta \in (0, 1)$, denoted by*

$$\sqrt{\rho} = \min\left\{\sqrt{\beta}\min\left\{\sqrt{\frac{\mu_f}{L_f}}, \sqrt{\frac{\mu_g}{L_g}}\right\}, (1 - \beta)\frac{\sqrt{\mu_f\mu_g}}{\|B\|}\right\}.$$

*Then for any $\alpha > 0$ and $\mathcal{E}$ defined by (55) and any two vectors $\mathbf{z}_k, \mathbf{z}_{k+1} \in \mathbb{R}^{2(m+n)}$*

$$\left(1 - \frac{\alpha}{\sqrt{\rho}}\right) D_\mathcal{E}(\mathbf{z}_k, \mathbf{z}_{k+1})$$
$$\leq D_\mathcal{E}(\mathbf{z}_k, \mathbf{z}_{k+1}) \pm \alpha\langle y_{k+1} - y_k, \nabla F(x_{k+1}) - \nabla F(x_k)\rangle \pm \alpha\|y_k - y_{k+1}\|^2_{\mathcal{B}^{\mathrm{sym}}}, \qquad (61)$$
$$\leq \left(1 + \frac{\alpha}{\sqrt{\rho}}\right) D_\mathcal{E}(\mathbf{z}_k, \mathbf{z}_{k+1}).$$

*In particular, for $\alpha \leq \sqrt{\rho}$, we have the bound*

$$0 \leq \mathcal{E}^\alpha(\mathbf{z}) \leq 2\mathcal{E}(\mathbf{z}) \quad \mathbf{z} \in \mathbb{R}^{2d}. \qquad (62)$$

*Proof.* For any $\beta \in (0,1)$, using the vector form of Lemma C.2, we have bound

$$\left(1 - \frac{\alpha}{\sqrt{\rho_F}}\right)\left(D_F(x_k, x_{k+1}) + \frac{\beta}{2}\|y_k - y_{k+1}\|_{\mathcal{D}_\mu}^2\right)$$

$$\leq D_F(x_k, x_{k+1}) + \frac{\beta}{2}\|y_k - y_{k+1}\|_{\mathcal{D}_\mu}^2 \pm \alpha\langle y_{k+1} - y_k, \nabla F(x_{k+1}) - \nabla F(x_k)\rangle \tag{63}$$

$$\leq \left(1 + \frac{\alpha}{\sqrt{\rho_F}}\right)\left(D_F(x_k, x_{k+1}) + \frac{\beta}{2}\|y_k - y_{k+1}\|_{\mathcal{D}_\mu}^2\right)$$

with $\rho_F = \beta \min\left\{\frac{\mu_f}{L_f}, \frac{\mu_g}{L_g}\right\}$.

Use Lemma G.1, we have the bound

$$\left(1 - \frac{2\alpha\|B\|}{(1-\beta)\sqrt{\mu_f\mu_g}}\right)\frac{1-\beta}{2}\|y_k - y_{k+1}\|_{\mathcal{D}_\mu}^2 \leq \frac{1-\beta}{2}\|y_k - y_{k+1}\|_{\mathcal{D}_\mu}^2 \pm \alpha\|y_k - y_{k+1}\|_{\mathcal{B}^{\text{sym}}}^2$$

$$\leq \left(1 + \frac{2\alpha\|B\|}{(1-\beta)\sqrt{\mu_f\mu_g}}\right)\frac{1-\beta}{2}\|y_k - y_{k+1}\|_{\mathcal{D}_\mu}^2. \tag{64}$$

Adding (63) and (64) implies (61).

Apply (61) with $\mathbf{z}_k = \mathbf{z}$ and $\mathbf{z}_{k+1} = \mathbf{z}^*$ to get (62). $\qquad\square$

In Lemma G.2, the optimal $\beta^*$ such that

$$\sqrt{\beta^*} \min\left\{\sqrt{\frac{\mu_f}{L_f}}, \sqrt{\frac{\mu_g}{L_g}}\right\} = (1 - \beta^*)\frac{\sqrt{\mu_f\mu_g}}{\|B\|}$$

gives the largest step size

$$\alpha^* = \sqrt{\rho^*} = \frac{\sqrt{\min\left\{\sqrt{\frac{\mu_f}{L_f}}, \sqrt{\frac{\mu_g}{L_g}}\right\}^2 + \frac{4\mu_f\mu_g}{\|B\|^2}} - \min\left\{\sqrt{\frac{\mu_f}{L_f}}, \sqrt{\frac{\mu_g}{L_g}}\right\}}{\frac{2\sqrt{\mu_f\mu_g}}{\|B\|}} \min\left\{\sqrt{\frac{\mu_f}{L_f}}, \sqrt{\frac{\mu_g}{L_g}}\right\}$$

$$= \left(1 - \frac{\left(\sqrt{\min\left\{\sqrt{\frac{\mu_f}{L_f}}, \sqrt{\frac{\mu_g}{L_g}}\right\}^2 + \frac{4\mu_f\mu_g}{\|B\|^2}} - \min\left\{\sqrt{\frac{\mu_f}{L_f}}, \sqrt{\frac{\mu_g}{L_g}}\right\}\right)^2}{\frac{4\mu_f\mu_g}{\|B\|^2}}\right)\frac{\sqrt{\mu_f\mu_g}}{\|B\|}$$

An alternative simple choice to check the order of convergence rate is to set $\beta = 3 - 2\sqrt{2}$ and the consequent step size is

$$\alpha = \sqrt{\rho} = (\sqrt{2} - 1)\min\left\{\sqrt{\frac{\mu_f}{L_f}}, \sqrt{\frac{\mu_g}{L_g}}, \frac{\sqrt{\mu_f\mu_g}}{\|B\|}\right\}.$$

Now we are ready to prove the theorem.

*Proof of Theorem 1.2.* We write (50) as a correction of the implicit Euler method

$$\mathbf{z}_{k+1} - \mathbf{z}_k = \alpha\mathcal{G}(\mathbf{z}_{k+1}) - \alpha\begin{pmatrix} 0 & I \\ \mathcal{D}_\mu^{-1}\nabla F & 0 \end{pmatrix}(\mathbf{z}_{k+1} - \mathbf{z}_k) - \alpha\begin{pmatrix} 0 \\ \mathcal{D}_\mu^{-1}\mathcal{B}^{\text{sym}}(y_{k+1} - y_k) \end{pmatrix}. \tag{65}$$

Use the definition of Bregman divergence, we have the identity for the difference of the Lyapunov function $\mathcal{E}$ at consecutive steps:

$$\mathcal{E}(\mathbf{z}_{k+1}) - \mathcal{E}(\mathbf{z}_k) = \langle\nabla\mathcal{E}(\mathbf{z}_{k+1}), \mathbf{z}_{k+1} - \mathbf{z}_k\rangle - D_\mathcal{E}(\mathbf{z}_k, \mathbf{z}_{k+1})$$

Substitute (65) and expand the cross term:

$$
\begin{aligned}
\langle \nabla \mathcal{E}(\mathbf{z}_{k+1}), \mathbf{z}_{k+1} - \mathbf{z}_k \rangle = {} & \alpha \langle \nabla \mathcal{E}(\mathbf{z}_{k+1}), \mathcal{G}(\mathbf{z}_{k+1}) \rangle \\
& - \alpha \langle \nabla F(x_{k+1}) - \nabla F(x^*), y_{k+1} - y_k \rangle - \alpha \langle y_{k+1} - x^*, \nabla F(x_{k+1}) - \nabla F(x_k) \rangle \\
& - \alpha \langle y_{k+1} - y^*, y_{k+1} - y_k \rangle_{\mathcal{B}^{\mathrm{sym}}}
\end{aligned}
\quad (66)
$$

We split the gradient term as

$$
\begin{aligned}
& \alpha \langle \nabla F(x_{k+1}) - \nabla F(x^*), y_{k+1} - y_k \rangle + \alpha \langle y_{k+1} - x^*, \nabla F(x_{k+1}) - \nabla F(x_k) \rangle \\
= {} & \alpha \langle \nabla F(x_{k+1}) - \nabla F(x^*), y_{k+1} - x^* \rangle + \alpha \langle y_{k+1} - y_k, \nabla F(x_{k+1}) - \nabla F(x_k) \rangle \\
& - \alpha \langle y_k - x^*, \nabla F(x_k) - \nabla F(x^*) \rangle.
\end{aligned}
$$

We can use identity of squares to expand

$$
\langle y_{k+1} - y^*, y_{k+1} - y_k \rangle_{\mathcal{B}^{\mathrm{sym}}} = \frac{1}{2} \left( \|y_{k+1} - y^*\|_{\mathcal{B}^{\mathrm{sym}}}^2 + \|y_{k+1} - y_k\|_{\mathcal{B}^{\mathrm{sym}}}^2 - \|y_k - y^*\|_{\mathcal{B}^{\mathrm{sym}}}^2 \right).
$$

Substitute back to (18) and rearrange terms, we obtain the following identity:

$$
\begin{aligned}
\mathcal{E}^\alpha(\mathbf{z}_{k+1}) - \mathcal{E}^\alpha(\mathbf{z}_k) = {} & \alpha \langle \nabla \mathcal{E}(\mathbf{z}_{k+1}), \mathcal{G}(\mathbf{z}_{k+1}) \rangle \\
& - D_{\mathcal{E}}(\mathbf{z}_k, \mathbf{z}_{k+1}) + \alpha \langle y_{k+1} - y_k, \nabla F(x_{k+1}) - \nabla F(x_k) \rangle - \alpha \|y_k - y_{k+1}\|_{\mathcal{B}^{\mathrm{sym}}}^2,
\end{aligned}
\quad (67)
$$

which holds for arbitrary step size $\alpha$.

Now take $\alpha \le \sqrt{\rho}$. The last term in (67) is non-positive. Use the strong Lyapunov property, and the bound $\mathcal{E}^\alpha(\mathbf{z}) \le 2\mathcal{E}(\mathbf{z})$ to get

$$
\mathcal{E}^\alpha(\mathbf{z}_{k+1}) - \mathcal{E}^\alpha(\mathbf{z}_k) \le -\alpha \mathcal{E}(\mathbf{z}_{k+1}) \le -\frac{\alpha}{2} \mathcal{E}^\alpha(\mathbf{z}_{k+1}),
\quad (68)
$$

which implies the global linear convergence:

$$
\mathcal{E}^\alpha(\mathbf{z}_{k+1}) \le \frac{1}{1 + \alpha/2} \mathcal{E}^\alpha(\mathbf{z}_k) \le \left( \frac{1}{1 + \alpha/2} \right)^{k+1} \mathcal{E}^\alpha(\mathbf{z}_0), \quad k \ge 0.
$$

Moreover, (68) implies

$$
\mathcal{E}(\mathbf{z}_{k+1}) \le \frac{1}{\alpha} (\mathcal{E}^\alpha(\mathbf{z}_k) - \mathcal{E}^\alpha(\mathbf{z}_{k+1})) \le \frac{1}{\alpha} \mathcal{E}^\alpha(\mathbf{z}_k) \le \frac{1}{\alpha} \left( \frac{1}{1 + \alpha/2} \right)^k \mathcal{E}^\alpha(\mathbf{z}_0),
$$

which coincides with (8) with $C_0 = \frac{1}{\alpha} \mathcal{E}^\alpha(\mathbf{z}_0) \ge 0$. $\qquad \square$

## H  AOR-HB-SADDLE-I ALGORITHM AND CONVERGENCE ANALYSIS

We propose the AOR-HB-saddle-I algorithm in the following Algorithm 4.

Consider the modified Lyapunov function

$$
\mathcal{E}^\alpha(\mathbf{z}) := \mathcal{E}(\mathbf{z}) + \alpha \langle \nabla F(x) - \nabla F(x^*), y - x^* \rangle,
\quad (69)
$$

The following theorem show the convergence rate of AOR-HB-saddle-I method.

**THEOREM H.1** (Convergence of AOR-HB-saddle-I method). *Suppose $f$ is $\mu_f$-strongly convex and $L_f$-smooth, $g$ is $\mu_g$-strongly convex and $L_g$-smooth and let $\sqrt{\rho} = \min\left\{ \sqrt{\frac{\mu_f}{L_f}}, \sqrt{\frac{\mu_g}{L_g}} \right\}$. Let $(u_k, v_k, p_k, q_k)$ be generated by Algorithm 4 with initial value $(u_0, v_0, p_0, q_0)$ and step size $\alpha = \sqrt{\rho}$. Then there exists a non-negative constant $C_0 = C_0(u_0, v_0, p_0, q_0, \mu_f, L_f, \mu_g, L_g)$ so that we have the linear convergence*

$$
D_f(u_{k+1}, u^*) + D_g(p_{k+1}, p^*) + \frac{\mu_f}{2} \|v_{k+1} - u^*\|^2 + \frac{\mu_g}{2} \|q_{k+1} - p^*\|^2 \le C_0 \left( \frac{1}{1 + \frac{1}{2}\sqrt{\rho}} \right)^k.
\quad (70)
$$

---

**Algorithm 4** AOR-HB-saddle-I Algorithm

---

1: **Parameters:** $u_0, v_0 \in \mathbb{R}^m, p_0, q_0 \in \mathbb{R}^n, L_f, L_g, \mu_f, \mu_g, \|B\|$.

2: **Set** $\alpha = \min\left\{ \sqrt{\frac{\mu_f}{L_f}}, \sqrt{\frac{\mu_g}{L_g}} \right\}$.

3: **for** $k = 0, 1, 2, \ldots$ **do**

4: $\quad u_{k+1} = \dfrac{1}{1+\alpha}(u_k + \alpha v_k); \quad p_{k+1} = \dfrac{1}{1+\alpha}(p_k + \alpha q_k)$

5: $\quad$ Find $(v_{k+1}, q_{k+1})$ such that

$$v_{k+1} = \frac{1}{1+\alpha}\left( v_k + \alpha u_{k+1} - \frac{\alpha}{\mu_f}\left(2\nabla f(u_{k+1}) - \nabla f(u_k) + B^\top q_{k+1}\right) \right),$$

$$q_{k+1} = \frac{1}{1+\alpha}\left( q_k + \alpha p_{k+1} - \frac{\alpha}{\mu_g}\left(2\nabla g(p_{k+1}) - \nabla g(p_k) - B v_{k+1}\right) \right).$$

6: **end for**

7: **return** $u_{k+1}, v_{k+1}, p_{k+1}, q_{k+1}$

---

*Proof.* AOR-HB-saddle-I can be written as a correction of the implicit Euler method

$$\mathbf{z}_{k+1} - \mathbf{z}_k = \alpha\mathcal{G}(\mathbf{z}_{k+1}) - \alpha \begin{pmatrix} 0 & I \\ \mathcal{D}_\mu^{-1}\nabla F & 0 \end{pmatrix} (\mathbf{z}_{k+1} - \mathbf{z}_k). \tag{71}$$

The proof follows the proof of Theorem 1.1 in Appendix C with $f = F$.

$\square$

