# OpenReview forum: "Accelerated Over-Relaxation Heavy-Ball Method: Achieving Global Accelerated Convergence with Broad Generalization"
_ICLR.cc/2025/Conference — ICLR 2025 Poster_

### Official Review · Reviewer_XLjC · 2024-10-23

**Soundness:** 3
**Presentation:** 2
**Contribution:** 3
**Rating:** 6
**Confidence:** 4

**Summary:**

This paper proposes a new momentum method AOR-HB. This method is motivated by discretizing an related ODE using over-relaxation. AOR-HB has accelerated rates compared to HB, and it is further extended to proximal and mini-max problems. Numerical results are provided.

**Strengths:**

1. Compared to HB, the proposed approach achieves accelerated rates.

2. The design of ODEs and discretizing can be useful.

**Weaknesses:**

1. The constant $C_0$ in theorem 1.1 still depends on $\kappa$, which implies that this approach remains slower than Nesterov's momentum. The author should make this dependence clear.

2. Extensions to convex and stochastic problems are less straightforward.

3. The proposed approach only outperforms HB in numerical results. This indicates that it is not the most efficient momentum methods from a practical standpoint.

In sum, while the theoretical finding are impressive, the resultant approach occupies a ‘mid-point’ between momentum methods both theoretically and empirically.

**Questions:**

See above.

---

> ### Author Response · Authors · 2024-11-19
>
> Thank you very much for your review. We are willing to discuss and improve the weakness points.
>
> - *The constant $C_0$ in theorem 1.1 still depends on $\kappa$, which implies that this approach remains slower than Nesterov's momentum. The author should make this dependence clear.*
>
> The observation that the constant $C_0$ in Theorem 1.1 still depends on $\kappa$ ia correct. In particular, we write
> $$
> C_0 = C_0(x_0, y_0,\mu, L) = \sqrt{\kappa}\ \mathcal E^{\alpha}(x_0, y_0)
> $$
> where
> $$
> \mathcal  E^{\alpha}(x, y):=D_f(x, x^*)+ \frac{\mu}{2}\\|y-x^{*}\\|^2 + \alpha  (\nabla f(x) - \nabla f(x^{\star}), x - x^{\star} ).
> $$
> We make this dependence clear in the revised paper and point out the constant can be eliminated if we take $\mathcal  E^{\alpha}(x, y)$ as the measure of convergence:
> $$
> \mathcal  E^{\alpha}(x_{k}, y_{k}) \leq \left (\frac{1}{1+\alpha/2} \right)^{k} \mathcal  E^{\alpha}(x_{0}, y_{0}).
> $$
> $\mathcal E^{\alpha}$ is a nonnegative quantity which implies a weaker congergence. The constant showed up when we use some inequalities to change back to $\mathcal E$ in order to control the strong convergence.
>
> ---
>
> - *Extensions to convex and stochastic problems are less straightforward.*
>
> AOR-HB methods can be extended to the convex setting ($\mu = 0$) while achieving accelerated convergence rates of $\mathcal{O}(1/k^2)$, comparable to NAG. For further details, please see the second point in the Global Rebuttal.
>
> Extending AOR-HB to stochastic problems is more challenging due to the introduction of sampling errors. While AOR-HB is primarily designed for deterministic settings, variance control becomes essential when dealing with sampled gradients. For instance, integrating [stochastic variance reduction](https://en.wikipedia.org/wiki/Stochastic_variance_reduction) techniques with AOR-HB could address this issue. We are currently investigating this direction and aim to gain deeper insights, and try to include Adam method within our framework.
>
>
>
>
> - *The proposed approach only outperforms HB in numerical results. This indicates that it is not the most efficient momentum methods from a practical standpoint.* *In sum, while the theoretical finding are impressive, the resultant approach occupies a ‘mid-point’ between momentum methods both theoretically and empirically.*
>
> For strongly convex optimization, AOR-HB numerically outperforms HB and ADR and is comparable to NAG. While Triple Momentum (TM) achieves the fastest convergence due to finely tuned parameters, accelerated methods with the same theoretical convergence rates (up to constants) tend to perform similarly asymptotically.
>
> **The primary strength of our approach lies in its generalization capability.** AOR-HB is particularly valuable for problems beyond convex minimization. For non-smooth and non-convex problems (Section 4.2), AOR-HB outperforms other accelerated methods. Additionally, AOR-HB shows superior efficiency in solving saddle-point problems (Section 4.3), making it a versatile and practical choice across diverse applications.
>
> Theoretically, AOR-HB's convergence proof readily extends to composite and saddle-point problems. In contrast, adapting other techniques, such as the estimated sequence method, to scenarios beyond the convex case involves significant effort.

---

> > ### Comment · Reviewer_XLjC · 2024-11-23
> >
> > Thanks for the responses. I tend to keep the score, for reasons below.
> >
> > - The $\kappa$ dependence of AOR-HB is less favorable compared to Nesterov's method.
> > - Even if other methods currently cannot be analyzed in certain settings, they may have a better $\kappa$ dependence. Hence, the generalization capability of AOR-HB comes with a tradeoff with convergence.

---

> ### Author Response · Authors · 2024-11-23
> **$\kappa$ dependence for the min-max problems**
>
> *Even if other methods currently cannot be analyzed in certain settings, they may have a better $\kappa$ dependence. Hence, the generalization capability of AOR-HB comes with a tradeoff with convergence.*
>
> Do you mean numerically or theoretically ?
>
> Numerically, AOR-HB converges similarly to Nesterov Accelerated Gradient (NAG) for convex optimization methods and **outperforms** other accelerated methods for min-max problems with bilinear coupling.
>
> Theoretically, we agree that the $\kappa$ dependence makes the convergence rate less favorable for convex optimization. However, in the min-max problem setting, for example, it is comparable to other results in:
>
> - Dmitry Kovalev, Alexander Gasnikov, and Peter Richtárik. *Accelerated primal-dual gradient method for smooth and convex-concave saddle-point problems with bilinear coupling.* Advances in Neural Information Processing Systems, 35:21725–21737, 2022.
> - Kiran K. Thekumparampil, Niao He, and Sewoong Oh. *Lifted primal-dual method for bilinearly coupled smooth minimax optimization.* In *International Conference on Artificial Intelligence and Statistics*, pp. 4281–4308. PMLR, 2022.
>
> For example, in LPD, the convergence result is
> $$
> \Delta(x, y)=\kappa_{x y}\left(\mu_x\left\|x-x^*\right\|^2+\mu_y\left\|y-y^*\right\|^2\right) .
> $$
>
>
> Theorem 2 (Informal, cf. Corollary 2). For any $k \geq$ 0 , set the parameters
> $$
> \gamma=1+\kappa^{-1}, \theta_k=1 / \gamma, \quad \eta_{x, k}=\left(\sqrt{\kappa_x-1}+2 \kappa_{x y}\right)^{-1} / \mu_x,
> $$
> $$
> \eta_{y, k}=\left(2 \kappa_{x y}+\sqrt{\kappa_y-1}\right)^{-1} / \mu_y \quad \eta_{u, k}=\left(\sqrt{\kappa_x-1}\right)^{-1}, \eta_{v, k}=\left(\sqrt{\kappa_y-1}\right)^{-1}.
> $$
>
> Then for any $K>0$, output of Algorithm 1 satisfies
> $$
> \Delta\left(x^K, y^K\right) \leq \exp \left(-\frac{(K-1)}{(\kappa+1)}\right)\left(\left(\frac{1}{\eta_{x, 0}}+\frac{L_x-\mu_x}{\eta_{u, 0}}\right)\left\|x^*-x_0\right\|^2+\left(\frac{1}{\eta_{y, 0}}+\frac{L_y-\mu_y}{\eta_{v, 0}}\right)\left\|y^*-y_0\right\|^2\right) .
> $$
> You will observe the $\kappa$ dependence for the squared distance of the error. In the work by Kovalev et al. (2022), the dependence is similar (it is too complicated to replicate here). We hypothesize that **such $\kappa$ dependence is theoretically unavoidable for the min-max problem**. In view of the Lyapunov function $E^{\alpha}$, this dependence is absent. See Appendix G, equation (68) in the latest revision of our paper.
>
> We summarize the findings in the following table and conclude that for the min-max problem, both numerically and theoretically, AOR-HB outperforms existing methods.
>
> |            | Theory                  | Numerics          |
> | ---------- | ----------------------- | ----------------- |
> | **Convex** | Slightly worse than NAG | Comparable to NAG |
> | **Saddle** | Comparable to others    | Best              |

---

### Official Review · Reviewer_C9is · 2024-11-03

**Soundness:** 2
**Presentation:** 3
**Contribution:** 2
**Rating:** 6
**Confidence:** 3

**Summary:**

The paper introduces Accelerated Over-Relaxation Heavy-Ball method which can converge at an accelerated rate for strongly-convex objective functions.

**Strengths:**

The authors provide proofs for the convergence of their AOR-HB method for smooth, strongly-convex objective function. The method is also proved to achieve convergence for a class of non-smooth optimization and min-max problems.

**Weaknesses:**

The AOR-HB method is very similar to Nesterov Acceleration (details in questions).

**Questions:**

Nesterov Acceleration (NAG) can be defined as $v_k = w_k + \beta (w_k - w_{k-1})$ and $w_{k+1} = v_k - \gamma \nabla f (v_k)$. We can eliminate term $w_k$ and achieve
$$
v_{k+1} = v_k - \gamma \nabla f (v_k) + \beta (v_k - v_{k-1}) - \gamma \beta [ \nabla f (v_k) - \nabla f (v_{k-1})]
$$
$$
v_{k+1} = v_k - \gamma \beta [ (1 + \frac{1}{\beta}) \nabla f (v_k) - \nabla f (v_{k-1})] + \beta (v_k - v_{k-1})
$$
So the only difference if 1 versus $\frac{1}{\beta}$ (where $\beta$ is close to 1) as I understand, can you elaborate more on the differences between these two methods and when will AOR-HB is more prefer compare to NAG. In fig 1 and 2 the empirical performance of AOR-HB and NAG are also very similar.

minor: line 373 typo double "that"

---

> ### Author Response · Authors · 2024-11-19
>
> Thank you for your deep insight and careful reading. We have correct the typo you pointed out in the revised paper.  We would like to answer you question on the differences between AOR-HB and NAG.
>
>
> - *Can you elaborate more on the differences between these two methods and when will AOR-HB is more prefer compare to NAG.*
>
> Your observation is indeed insightful, and we deeply appreciate it. To ensure clarity for all reviewers, let us first restate your detailed formulation.
>
> When eliminating certain variables, NAG takes the form:
> $$
> v_{k+1} = v_k -\gamma \beta \left[ \left( 1 + \frac{1}{\beta}\right) \nabla f(v_k) - \nabla f(v_{k-1}) \right]  + \beta (v_{k} - v_{k-1}),
> $$
> where, for strongly convex minimization, the parameters are set as:
> $$
> \gamma = \frac{1}{L}, \quad \beta = \frac{\sqrt{L} - \sqrt{\mu}}{\sqrt{L} + \sqrt{\mu}}.
> $$
>
> In contrast, AOR-HB takes the form:
> $$
> v_{k+1} = v_k -\gamma \beta \left(2 \nabla f(v_k) - \nabla f(v_{k-1}) \right)  + \beta (v_{k} - v_{k-1}),
> $$
> with parameters:
> $$
> \gamma = \frac{1}{L}, \quad \beta = \frac{L}{(\sqrt{L} + \sqrt{\mu})^2}.
> $$
>
> While the parameter choices differ, as $L \gg \mu$ (i.e., $\kappa \gg 1$), AOR-HB and NAG asymptotically converge to the same form when $\kappa \to +\infty$. In this sense, AOR-HB can be interpreted as an $\mathcal{O}(1/\kappa)$ perturbation of NAG. This likely accounts for the observed numerical similarity in their performance on ill-conditioned strongly convex problems.
>
> For convex optimization ($\mu = 0$), AOR-HB also acts as a perturbation of NAG, with the algorithm:
> $$
> x_{k+1} = x_k - \frac{k}{k+3} \frac{1}{L} \big(2\nabla f(x_k) - \nabla f(x_{k-1})\big) + \frac{k}{k+3} (x_k - x_{k-1}).
> $$
> This formulation closely resembles NAG. Numerically (though due to page limits, examples are not included in the revision), we observe similar performance between AOR-HB and NAG for non-strongly convex problems, reinforcing their close relationship in these settings.
>
> ---
>
> Thank you for raising this important point. We have added a short paragraph in the introduction (below Theorem 1.1) addressing this. We view the close relationship between AOR-HB and NAG as a strength of our approach. For strongly convex optimization, AOR-HB provides an alternative explanation for the acceleration mechanism of NAG. Moreover, this mechanism is extendable to other scenarios, whereas the estimated sequence method used for NAG is notably difficult to generalize.
>
> This highlights AOR-HB’s key advantage: **superior generalization capability**. For further details, please refer to the first point in the Global Rebuttal.

---

> > ### Author Response · Authors · 2024-11-25
> > **Comparison with NAG**
> >
> > We kindly remind the reviewer to review our rebuttal. For your convenience, we have summarized the findings in the table below. For more details on the generalization capability, please refer to the first point in the Global Rebuttal.
> >
> > |            | Theory                  | Numerics          |
> > | ---------- | ----------------------- | ----------------- |
> > | **Convex** | Slightly worse than NAG | Comparable to NAG |
> > | **Saddle** | Comparable to others    | Best              |

---

> > > ### Comment · Reviewer_C9is · 2024-11-25
> > >
> > > Thank you for your response. It had addressed my question and concern. I changed the score to 6.

---

> > > > ### Author Response · Authors · 2024-11-25
> > > >
> > > > Thank you for raising the score and for your insightful observations on the relation to NAG. As noted by Reviewer 1fRV, AOR-HB can also be viewed as a higher-order perturbation of the lifted primal-dual (LPD) method. For strongly convex optimization problems, all accelerated first-order methods are, in essence, equivalent up to higher-order perturbations of parameters.
> > > >
> > > > However, what sets AOR-HB apart is its superior generalization ability beyond the convex optimization regime. We believe this distinction highlights the broader potential of AOR-HB and its contribution to the field.
> > > >
> > > > Once again, we sincerely appreciate your thoughtful feedback and support in improving our work.

---

### Official Review · Reviewer_YQdx · 2024-11-03

**Soundness:** 4
**Presentation:** 4
**Contribution:** 3
**Rating:** 6
**Confidence:** 3

**Summary:**

This paper proposes a variant of heavy-ball momentum acceleration that provably achieves the optimal linear convergence rate with the factor of $\sqrt{\kappa}$ for smooth strongly convex functions, amending the previous heavy-ball momentum method that does not accelerate in this setting. The proposed framework is further extended to composite optimization and saddle point optimization. Numerical results are provided to validate the theory.

**Strengths:**

1. The proposed method improves heavy-ball momentum, a well-known and widely applied method, to achieve acceleration for strongly convex functions.
2. Comprehensive extensions to composite optimization and saddle point optimization are also included.

**Weaknesses:**

1. The paper can be further improved if the authors could include some more intuitive discussion on why the over-relaxation is applied to approximate the gradient in line 110.

**Questions:**

1. How are the convergence rate and practical performance of AOR-HB for non-strongly convex functions? Does the proposed modification also achieve the optimal $O(1/T^2)$ rate or is the method limited to strongly convex functions?
2. In Figure 2, the HB method also achieves fast linear convergence, faster than all the other methods. Why is it the case and wouldn't this weaken the motivation of the paper? Are there more empirical evidence to demonstrate the contrast between HB and AOR-HB in terms of optimizing strongly convex functions?

---

> ### Author Response · Authors · 2024-11-19
>
> We appriciate your time and effort to review our paper and thank you for giving positive referring and constructive comments. We would like to address the questions in details.
>
> ---
>
> **Weaknesses:**
>
> - *The paper can be further improved if the authors could include some more intuitive discussion on why the over-relaxation is applied to approximate the gradient in line 110.*
>
>
> The motivation of using AOR techniques: $\nabla f(x) \approx 2\nabla f(x_k) - \nabla f(x_{k+1})$ is to symmetrize the error equation, cf. (19) in the revised paper, where we attempt to illustrate this idea using a maxtrix version with a quadratic objective function. With such symmetrization, we can use identity of squares. The square terms from this identity are absorbed into the Lyapunov function, resulting in an **equality** rather than an inequality.
>
> We add a sentence in the revised paper in line 121.
>
> ---
>
> **Questions:**
>
> - *How are the convergence rate and practical performance of AOR-HB for non-strongly convex functions? Does the proposed modification also achieve the optimal $O(1/T^2)$ rate or is the method limited to strongly convex functions?*
>
> Our approach extends to the non-strongly convex setting ($\mu = 0$) using the dynamic scaling technique introduced by Luo & Chen (2022), which achieves an accelerated rate of $\mathcal{O}(1/k^2)$, comparable to NAG. Please refer to the second point in the Global Rebuttal for further details.
>
>
>
> - *In Figure 2, the HB method also achieves fast linear convergence, faster than all the other methods. Why is it the case and wouldn't this weaken the motivation of the paper? Are there more empirical evidence to demonstrate the contrast between HB and AOR-HB in terms of optimizing strongly convex functions?*
>
> Convergence analysis provides worst-case rates for algorithms over a class of problems. While HB performs well on specific instances, as seen in Figure 2, its fast convergence lacks theoretical guarantees and may fail in cases like Figure 1. This highlights the importance of robust theoretical analysis over empirical success alone.
>
> Lessard et al. (2016, Fig. 6 and 7, p. 81) demonstrated a non-convergent example, showing that Polyak's HB method does not guarantee global convergence for general strongly convex problems. In contrast, AOR-HB is the first HB-type method with provable global convergence and an accelerated rate. Here, "global" means convergence from any starting point, whereas Polyak's HB converges quickly only when sufficiently close to the minimum—a condition that is impractical to verify in practice.
>
> Moreover, the absence of acceleration in HB is not merely a technical gap. Goujaud et al. (2023) recently proved that HB fails to achieve accelerated rates for smooth and strongly convex problems, with additional empirical evidence provided in their work.
>
> ---
>
> ### References
> - Hao Luo and Long Chen. From differential equation solvers to accelerated first-order methods for convex optimization. *Mathematical Programming*, 195(1):735–781, 2022.
> - Laurent Lessard, Benjamin Recht, and Andrew Packard. Analysis and design of optimization algo- rithms via integral quadratic constraints. *SIAM Journal on Optimization*, 26(1):57–95, 2016.
> - Baptiste Goujaud, Adrien Taylor, and Aymeric Dieuleveut. Provable non-accelerations of the heavy- ball method. *arXiv preprint arXiv:2307.11291*, 2023.

---

> > ### Comment · Reviewer_YQdx · 2024-11-23
> >
> > Thank the authors for the reply. It addresses my concerns and I'm willing to keep my score.

---

### Official Review · Reviewer_1fRV · 2024-11-06

**Soundness:** 3
**Presentation:** 2
**Contribution:** 2
**Rating:** 6
**Confidence:** 3

**Summary:**

This paper proposes a variant of the classical heavy-ball method, where the key modification involves replacing $\nabla f(x_k)$ with its over-relaxation $2 \nabla f(x_k) - \nabla f(x_{k-1})$. It can be regarded as a discretization of the rotated gradient flow introduced by (Chen & Luo, 2021). By employing a Lyapunov function, the authors prove that the proposed method achieves the optimal linear convergence rate for smooth strongly convex minimization problems. They further extended their results to the composite setting and strongly-convex-strongly-concave saddle point problems with bilinear coupling.

**Strengths:**

- It is known that the classical heavy-ball method cannot achieve the accelerated rate for general smooth strongly-convex optimization problems. The contribution of this paper is to show that, with a simple modification, a heavy-ball-like method can indeed achieve acceleration.
- The proof technique is conceptually simpler than the standard approach used in accelerated gradient methods, such as the estimate sequence. Furthermore, this technique is adaptable to more general settings, as evidenced by the extensions provided in the paper.

**Weaknesses:**

- The main drawback is that other acceleration methods, such as Nesterov's accelerated gradient (NAG) and its extensions, are known to achieve similar convergence guarantees in the considered settings. As such, the advantage of the proposed algorithm over these existing methods remains unclear. This is further illustrated in the numerical experiments, where the proposed method exhibits comparable performance to NAG. Therefore, I have reservations about referring to the result as a "breakthrough."
- Another limitation is that the paper focuses solely on the strongly convex setting. It is unclear if an accelerated rate can also be achieved in the more general convex setting.

**Questions:**

- It appears that the proposed method resembles the accelerated method proposed by Thekumparampil et al. (2022), albeit from a different perspective. Specifically, they presented an accelerated method for strongly convex minimization problems in equation (11). If the variable $x_k$ is eliminated, the resulting update rule resembles
$$x_{k+1} = x_k - \gamma (\nabla f(x_k) - \theta \nabla f(x_{k-1}))+ \beta (x_k - x_{k-1}),$$
where $\gamma,\theta,\beta$ are some constants. A detailed comparison will be helpful.

-The notation of $\mathbf{x} = (x,y)^\top$ could be confusing. I suggest the authors use a different symbol, such as $\mathbf{z}$, to represent the concatenation of $x$ and $y$.

----
Thekumparampil, Kiran K., Niao He, and Sewoong Oh. "Lifted primal-dual method for bilinearly coupled smooth minimax optimization." AISTATS 2022.

---

> ### Author Response · Authors · 2024-11-19
>
> Thank you for your thoughtful review and for recognizing the simplicity of our proofs compared to other approaches in accelerated gradient methods. We appreciate your insightful questions and constructive feedback. Below, we address your concerns and clarify the points you raised.
>
> ---
>
> ### Weaknesses
>
> - *The advantage of the proposed algorithm over these existing methods remains unclear. …  Therefore, I have reservations about referring to the result as a "breakthrough."*
>
> Thank you for your comments. In response, we have removed the word "breakthrough" to maintain a more modest tone. We greatly appreciate the constructive feedback and have revised the introduction to better emphasize the **superior generalization capability** of our approach. For further details, please refer to the first point in the Global Rebuttal.
>
> - *Another limitation is that the paper focuses solely on the strongly convex setting. It is unclear if an accelerated rate can also be achieved in the more general convex setting.*
>
> Our approach extends to the non-strongly convex setting ($\mu = 0$), which achieves an accelerated rate of $\mathcal{O}(1/k^2)$, comparable to NAG. Please refer to the second point in the Global Rebuttal for further details.
>
>
> ---
>
> ### Questions
>
> - *The notation of $\mathbf{x} = (x,y)^\top$ could be confusing. I suggest the authors use a different symbol, such as $\mathbf{z}$, to represent the concatenation of $x$ and $y$.*
>
> Thanks for the nice suggestion. We have changed $\mathbf x$ to $\mathbf z$.
>
> We will discuss the difference between AOR-HB and LPD in a separated reply.

---

> ### Author Response · Authors · 2024-11-19
> **AOR-HB and LPD**
>
> - *It appears that the proposed method resembles the accelerated method proposed by Thekumparampil et al. (2022), albeit from a different perspective. Specifically, they presented an accelerated method for strongly convex minimization problems in equation (11). If the variable $x_k$ is eliminated, the resulting update rule resembles $$x_{k+1} = x_k - \gamma (\nabla f(x_k) - \theta \nabla f(x_{k-1}))+ \beta (x_k - x_{k-1}),$$ where $\gamma,\theta,\beta$ are some constants. A detailed comparison will be helpful.*
>
> This is a very interesting question! We shall discuss the connection and difference between AOR-HB and LPD in details.
>
> Regarding to the LPD for strongly convex minimization problems in Thekumparampil, He, and Oh (2022), equation (11), we  eliminate $x_k$  to get a update rule for $\underline{x}_k$.
>
> Due to the markdown rendering issues on the OpenReview platform, we will continue using $x_k$ in the following formulas:
> $$
> (1+\eta_x \mu) \big [(1+\eta_u) x_{k+1} - x_{k}\big ] =  (1+\eta_u )x_k - x_{k-1} - \eta_u \eta_x [(1+\theta) \nabla \underline{f}(x_k) - \theta  \nabla \underline{f}(x_{k-1}) ].
> $$
> Rearraging terms and using the relation $\eta_x\mu = \eta_u$ and $\theta = 1/(1+\eta_u)$, this is equivalent to
> $$
> x_{k+1} = \left(1 -  \frac{\eta_u^2}{(1+\eta_u)^2}\right ) x_{k} - \frac{\eta_u^2 (2+\eta_u)}{\mu (1+\eta_u)^3} \left ( \nabla \underline{f}(x_k) - \frac{1}{2+\eta_u}  \nabla \underline{f}(x_{k-1}) \right) + \frac{1}{(1+\eta_u)^2}(x_{k} - x_{k-1}).
> $$
>
> Noticed that $\nabla \underline f(x) = \nabla f(x) - \mu x $, we can further formulate LPD update as
> $$
> x_{k+1} =~ \left(1 -  \frac{\eta_u^2}{(1+\eta_u)^2}\right ) x_{k} - \frac{\eta_u^2 (2+\eta_u)}{\mu (1+\eta_u)^3} \left ( \nabla f(x_k) - \frac{1}{2+\eta_u}  \nabla f(x_{k-1}) \right)
> $$
> $$
> \quad + \frac{\eta_u^2 (2+\eta_u)}{(1+\eta_u)^3}\left ( x_k - \frac{1}{2+\eta_u} x_{k-1} \right)+ \frac{1}{(1+\eta_u)^2}(x_{k} - x_{k-1}).
> $$
> If we assume that the considered problem is ill-conditioned, the parameter $\eta_u = \sqrt{\underline{\rho}}:= \sqrt{1/(\kappa -1)} \ll 1$ . We can further simplify
> $$
> \begin{aligned}
> x_{k+1} =&~ \left(1 -  \frac{\underline{\rho}}{(1+\sqrt{\underline{\rho}})^2}\right ) x_{k} - \frac{1}{L (1+\sqrt{\underline{\rho}})^2} \left ( \frac{2+\sqrt{\underline{\rho}}}{1+\sqrt{\underline{\rho}}}\nabla f(x_k) - \frac{1}{1+\sqrt{\underline{\rho}}}  \nabla f(x_{k-1}) \right)
> \end{aligned}
> $$
> $$
> \quad + \frac{\underline{\rho}}{(1+\sqrt{\underline{\rho}})^2}\left ( \frac{2+\sqrt{\underline{\rho}}}{1+\sqrt{\underline{\rho}}} x_k - \frac{1}{1+\sqrt{\underline{\rho}}} x_{k-1} \right)+ \frac{1}{(1+\sqrt{\underline{\rho}})^2}(x_{k} - x_{k-1})
> $$
> $$
> = \quad ~ \left(1 -  \frac{\underline{\rho}}{(1+\sqrt{\underline{\rho}})^2}\right ) x_{k} - \frac{1}{L (1+\sqrt{\underline{\rho}})^2} \left ( \left(2 - \frac{\sqrt{\underline{\rho}}}{1+\sqrt{\underline{\rho}}}\right)\nabla f(x_k) -\left (1 - \frac{\sqrt{\underline{\rho}}}{1+\sqrt{\underline{\rho}}} \right)  \nabla f(x_{k-1}) \right)
> $$
> $$
> \quad + \frac{\underline{\rho}}{(1+\sqrt{\underline{\rho}})^2}\left ( \frac{2+\sqrt{\underline{\rho}}}{1+\sqrt{\underline{\rho}}} x_k - \frac{1}{1+\sqrt{\underline{\rho}}} x_{k-1} \right)+ \frac{1}{(1+\sqrt{\underline{\rho}})^2}(x_{k} - x_{k-1}).
> $$
>
> Now, if we omit the higher order  $\mathcal O(\sqrt{\underline{\rho}})$ - terms, we resemble the AOR-HB type method
> $$
> \begin{aligned}
> x_{k+1} =&~ x_{k} - \frac{1}{L (1+\sqrt{\underline{\rho}})^2} \left ( 2\nabla f(x_k) -   \nabla f(x_{k-1}) \right) + \frac{1}{(1+\sqrt{\underline{\rho}})^2}(x_{k} - x_{k-1}).
> \end{aligned}
> $$
>
>
> From this perspective, for strongly convex optimization, AOR-HB simplifies the LPD method by reducing higher-order terms while still achieving the same accelerated convergence rates.
>
> The LPD method proposed by Thekumparampil et al. (2022) is designed for min-max problems with bilinear coupling. Both AOR-HB-saddle and LPD are single-loop algorithms that require similar computational effort, such as gradient evaluations and matrix-vector products. However, LPD involves five parameters: $\\{ \eta_{x, k}, \eta_{y, k}, \eta_{u, k}, \eta_{v, k}, \theta_k  \\}_{k=0}^{K-1}$, whereas AOR-HB-saddle requires only a single parameter: the step size $\alpha$ for time discretization. According to Thekumparampil et al. (2022) and our Theorem 1.2, both methods achieve accelerated convergence rate with optimal complexity.
>
> We are actively working to implement the LPD method (Algorithm 1 in Thekumparampil et al. (2022)) and conduct numerical comparisons.

---

> > ### Author Response · Authors · 2024-11-20
> > **Numerical comparison with LPD**
> >
> > Thanks to Thekumparampil’s help, we have included the lifted primal-dual method in our numerical comparisons. See the updated Figures 5 and 6. Our conclusion remains consistent: "our AOR-HB-saddle methods consistently exhibit faster linear convergence compared to other algorithms."

---

> > > ### Comment · Reviewer_1fRV · 2024-11-24
> > > **Follow-up questions**
> > >
> > > I thank the authors for addressing my concerns and performing additional numerical results. However, I have two follow-up questions and would appreciate your clarification.
> > >
> > > - In the non-strongly convex setting, the authors propose a modified AOR-HB method with time-varying parameters $\gamma$ and $\beta$. They claim that this method achieves a convergence rate of $O(1/k^2)$ using a perturbation argument in (Lessard et al., 2016). However, the approach in (Lessard et al., 2016) involves applying a linearly convergent method to a perturbed problem of the form $f(x) + \frac{\epsilon}{2}\\|x\\|^2$. It is unclear to me how this approach would lead to the update rule proposed by the authors, especially the choice of the coefficient $\frac{k}{k+3}$. The authors should provide more details if they would like to include this in the revision.
> > > - The main machinery used in this paper, including the rotated gradient flow and strong Lyapunov property, has been previously introduced to design accelerated methods (Chen & Luo, 2021; Luo & Chen, 2022). Could the author briefly comment on what sets this paper apart from these prior works?

---

> > > > ### Author Response · Authors · 2024-11-25
> > > > **The non-strongly convex setting**
> > > >
> > > > - *In the non-strongly convex setting, the authors propose a modified AOR-HB method with time-varying parameters $\gamma$ and $\beta$. They claim that this method achieves a convergence rate of $O\left(1 / k^2\right)$ using a perturbation argument in (Lessard et al., 2016). However, the approach in (Lessard et al., 2016) involves applying a linearly convergent method to a perturbed problem of the form $f(x)+\frac{\epsilon}{2}\|x\|^2$. It is unclear to me how this approach would lead to the update rule proposed by the authors, especially the choice of the coefficient $\frac{k}{k+3}$. The authors should provide more details if they would like to include this in the revision.*
> > > >
> > > > ---
> > > >
> > > > We thank the reviewer for the insightful comment. You are absolutely correct that the perturbation argument cannot derive the dynamic coefficient. To address the $\mu = 0$ case, we propose two approaches and have revised our remarks accordingly:
> > > >
> > > > 1. When $\mu = 0$, we propose a variant of the AOR-HB method that incorporates dynamic time rescaling, achieving an accelerated rate of $\mathcal O(1/k^2)$, comparable to NAG. The detailed proofs for this approach are provided in Appendix D of the revised version, building upon the framework proposed by Chen and Luo (2021).
> > > >
> > > > 2. Another approach involves extending the acceleration by considering the perturbed objective $f(x) + \frac{\epsilon}{2}\|x\|^2$, as discussed in  Lessard et al. (2016, Section 5.4).
> > > >
> > > > We have included the detailed convergence proof for the convex optimization in a new appendix (Appendix D).
> > > >
> > > > For saddle point problems, we have also added a remark on page 8 to address this case. Specifically, we reference the work of Thekumparampil et al. (2022), which impressively handles the $\mu = 0$ case for saddle point problems. In our work, we focus on the $\mu > 0$ setting but note the following:
> > > >
> > > > When $\mu_f = 0$ and $\mu_g > 0$, AOR-HB-saddle can be extended to the perturbed problem $\mathcal L(u,p) + \frac{\lambda \epsilon}{2} \|u\|^2$, achieving a convergence rate $\mathcal{O}\left(\sqrt{L_f / \varepsilon}+\|B\| / \sqrt{\mu_g \varepsilon}+\sqrt{L_g / \mu_g}\right) \log (|\epsilon|)$, which is optimal up to logarithmic factors. For more discussion with respect to using the perturbation argument, we refer to Thekumparampil et al. (2022)  and references therein.
> > > >
> > > > We hope these additions address your concerns, and we sincerely thank you for helping us improve the clarity and scope of the paper.

---

> > > > ### Author Response · Authors · 2024-11-26
> > > >
> > > > As the deadline for the final revision of the paper approaches, we wanted to kindly check if our replies address your follow-up questions satisfactorily. Once again, we deeply appreciate your thoughtful questions and comments, which have significantly improved the quality and clarity of our work.

---

> > > > > ### Comment · Reviewer_1fRV · 2024-11-28
> > > > >
> > > > > I thank the authors for addressing my concerns and have accordingly raised my score. I strongly encourage the authors to incorporate a comparison with (Chen & Luo, 2021; Luo & Chen, 2022) in the revised version as well.

---

> ### Author Response · Authors · 2024-11-24
> **New contribution**
>
> Thank you for the thoughtful questions. Regarding the first question, I’ll need to consult with my collaborator, who is currently in China, where it is midnight. I’ll get back to you on that.
>
> For now, let me focus on addressing the second question in this reply.
>
> *The main machinery used in this paper, including the rotated gradient flow and strong Lyapunov property, has been previously introduced to design accelerated methods (Chen & Luo, 2021; Luo & Chen, 2022). Could the author briefly comment on what sets this paper apart from these prior works?*
>
> Yes, while the algorithmic design and convergence analysis in this work follow the framework of the rotated gradient flow (for algorithm design) and strong Lyapunov property (for convergence analysis), our paper introduces significant contributions that set it apart from prior works.
>
>
> **Continuous Level**
>
> In the framework of Chen and Luo, the rotated gradient flow is represented as:
> $$
> \begin{bmatrix}  - I & \quad I \\\
> I - \mu^{-1} \nabla f & \quad - I
> \end{bmatrix}.
> $$
> In contrast, our work introduces an additional term, resulting in the flow:
> $$
> \begin{bmatrix}  - I & \quad I \\\
> I - \mu^{-1} \nabla f & \quad - I   - \mu^{-1} N
> \end{bmatrix}.
> $$
> The key contribution lies in the introduction of the $N$ term. Importantly, the strong Lyapunov property still holds for a broad class of $N$, significantly enhancing the generalization capability of the method.
>
> For example, in composite convex optimization, the comparison is as follows:
> $$
> \begin{bmatrix}  - I & \quad I \\\
> I - \mu^{-1} (\nabla f + \partial g) & \quad - I
> \end{bmatrix}
> \quad \text{vs.} \quad
> \begin{bmatrix}  - I & \quad I \\\
> I - \mu^{-1} \nabla f & \quad - I - \mu^{-1} \partial g
> \end{bmatrix}.
> $$
> More importantly, by selecting $N(x) = \begin{pmatrix} 0 & B^{\top} \\\ B & 0 \end{pmatrix}$, our approach extends to the min-max problem, which the original rotated gradient flow cannot handle.
>
>
> **Discrete Level**
>
> On the discrete level, we introduce the AOR (Accelerated Over-Relaxation) technique to approximate the gradient term as:   $$\nabla f(x) \approx 2\nabla f(x_{k+1}) - \nabla f(x_k),$$
> whereas Chen and Luo use a simpler discretization, $\nabla f(x) \approx \nabla f(x_{k+1})$, and require an additional gradient step or carefully designed extrapolation scheme to achieve acceleration.
>
> The AOR discretization is more faithful to the structure of the continuous flow. This fidelity not only simplifies convergence analysis but also significantly enhances generalization capabilities. For instance, beyond gradient terms, AOR can also be applied to bilinear couplings, such as   $$Bv \approx B (2v_{k+1} - v_k),$$
>
> resulting in accelerated first-order methods for min-max problems with bilinear coupling.

---

> ### Author Response · Authors · 2024-11-28
>
> Thank you for raising the score. We have incorporated a comparison with the approaches of Chen & Luo.
>
> - On page 3, we revised the text to:
>   *The AOR is used to symmetrize the error equation (19), representing a novel contribution compared to Luo & Chen (2022) and Chen & Luo (2021).*
>
> - On page 4, we revised the text to:
>
>   *Strongly convex optimization: $A(x) = \nabla f(x), \ N = 0$. This corresponds to the rotated flow developed in Luo & Chen (2022) and Chen & Luo (2021).*

---

### Author Response · Authors · 2024-11-19
**Global Rebuttal**

We thank all the reviewers for carefully reading our work. We appreciate your insightful questions and constructive feedback. The manuscript has been revised to address these comments. To meet the page limit, we have slightly shortened the abstract.

----

## Two primary concerns

The two primary concerns are 1) the advantages of AOR-HB over existing acceleration methods and 2) its extension to the convex case. Below, we provide a detailed response to these concerns.


---

**The advantage of the proposed algorithm compared to existing methods, particularly the Nesterov Accelerated Gradient (NAG) method**

For strongly convex optimization, acceleration can be viewed from various perspectives. AOR-HB can be interpreted as a parameter perturbation of NAG method, as noted by reviewer C9is. The convergence and performance of AOR-HB are thus comparable to those of NAG, though less precise than the triple momentum (TM) methods in the case of strongly convex optimization.

However, extending methods like NAG, TM, or high-resolution ODEs beyond convex optimization remains both rare and challenging. In contrast, AOR-HB demonstrates **superior generalization capability** as the true driving force of acceleration is through the $2 \times 2$ first-order ODE model.

We give a unified framework for convex optimization, saddle-point problems, and monotone operator equations, represented as:
$$\begin{bmatrix}  x'\\\  y'  \end{bmatrix}  =  \begin{bmatrix}    - I &  I \\\    I - \mu^{-1} A & - I - \mu^{-1} N  \end{bmatrix}  \begin{bmatrix}  x\\\  y  \end{bmatrix}.$$
- *Strongly convex optimization*: $A(x) = \nabla f(x), \ N = 0$.
- *Composite convex optimization:* $A(x) = \nabla f(x), \  N(y) = \partial g(y)$.
- *Saddle-point problem with bilinear coupling:*
  $A(x) = \begin{bmatrix}\nabla f(u) & 0 \\\ 0 & \nabla g(p)\end{bmatrix}, \ N = \begin{bmatrix}0 & B^{\top} \\\ B & 0 \end{bmatrix}$.
- *Monotone operators*: $A(x) = \nabla F(x), \ N$ is linear and skew-symmetric.

Simplicity in design enables extension to broader problems, while methods like NAG, TM, and high-resolution ODEs are tailored for convex objectives, limiting their flexibility. For example, adapting Nesterov’s method to composite problems requires significant effort.

For saddle-point problems, our AOR-HB-saddle achieves accuracy in fewer iterations compared to other algorithms. Its single-loop structure and minimal parameter tuning also enhance practicality and ease of implementation.

In summary, **the primary strength of our approach lies in its generalization capability.**


---

**Another limitation is that the paper focuses solely on the strongly convex setting. It is unclear if an accelerated rate can also be achieved in the more general convex setting.**

Our approach extends to the non-strongly convex setting ($\mu = 0$) using the dynamic scaling technique introduced by Luo & Chen (2022). The resulting algorithm is:
$$
x_{k+1} = x_k - \frac{k}{k+3} \frac{1}{L} \big(2\nabla f(x_k) - \nabla f(x_{k-1})\big) + \frac{k}{k+3} (x_k - x_{k-1}),
$$
which achieves an accelerated rate of $\mathcal{O}(1/k^2)$, comparable to NAG. Another approach involves using the perturbation form $f(x) + \frac{\epsilon}{2}\|x\|^2$, as discussed in Lessard et al. (2016, Section 5.4).

The perturbation approach also extends to composite convex optimization and saddle-point problems. However, due to page limits, we focused on the simplest case to present the core ideas clearly.

We have added a remark in the revision, immediately after Theorem 1.1 and Appendix D for the detailed proof.





---

## Major Improvement

Theorem 1.2 has been modified to allow a larger step size for AOR-HB-saddle which further speeds up the convergence of our new methods.

We are grateful to Reviewer 1fRV for highlighting the lifted primal-dual (LPD) method proposed by Thekumparampil et al. (2022), which is tailored for min-max problems with bilinear coupling. With Thekumparampil’s help, we have included the LPD in our numerical comparisons. See the updated Figures 5 and 6. Our conclusion remains unchanged: "our AOR-HB-saddle methods consistently exhibit faster linear convergence compared to other algorithms."

---

**Reference**
- Long Chen and Hao Luo. A unified convergence analysis of first order convex optimization methods via strong Lyapunov functions. *arXiv preprint arXiv:2108.00132*, 4(9):10, 2021.
- Hao Luo and Long Chen. From differential equation solvers to accelerated first-order methods for convex optimization. *Mathematical Programming*, 195(1):735–781, 2022.
- Laurent Lessard, Benjamin Recht, and Andrew Packard. Analysis and design of optimization algorithms via integral quadratic constraints. *SIAM Journal on Optimization*, 26(1):57–95, 2016.
- Thekumparampil, Kiran K., Niao He, and Sewoong Oh. *Lifted primal-dual method for bilinearly coupled smooth minimax optimization.* International Conference on Artificial Intelligence and Statistics. PMLR, 2022.

---

> ### Author Response · Authors · 2024-11-22
> **Feedback**
>
> We kindly remind the reviewers to read our rebuttal and share your thoughts with us. Once again, we deeply appreciate your constructive feedback, which has greatly improved both the results and the presentation of our work. Your insightful comments have been a valuable learning experience for us, and we thank you for your time and effort.

---

> > ### Author Response · Authors · 2024-11-27
> > **Reminder**
> >
> > * November 27th: Last day that authors may upload a revised PDF. After this date, authors may only post replies on the forum (no change).
> >
> > As the deadline for the final revision of our paper approaches, we wanted to kindly check if our responses have addressed your questions and comments satisfactorily. Your thoughtful feedback has been invaluable in enhancing the quality and clarity of our work, and we are deeply grateful for your time and insights.
> >
> > Wishing you a Happy Thanksgiving!

---

### Author Response · Authors · 2024-11-28
**New contribution compare to Chen and Luo's approach**

Reviewer 1fRV raised a follow-up question, and we believe it is important to include our response in the global rebuttal section to clarify our new contribution. This clarification has also been incorporated into the revised manuscript.

*The main machinery used in this paper, including the rotated gradient flow and strong Lyapunov property, has been previously introduced to design accelerated methods (Chen & Luo, 2021; Luo & Chen, 2022). Could the author briefly comment on what sets this paper apart from these prior works?*

Yes, while the algorithmic design and convergence analysis in this work follow the framework of the rotated gradient flow (for algorithm design) and strong Lyapunov property (for convergence analysis), our paper introduces significant contributions that set it apart from prior works.


**Continuous Level**

In the framework of Chen and Luo, the rotated gradient flow is represented as:
$$
\begin{bmatrix}  - I & \quad I \\\
I - \mu^{-1} \nabla f & \quad - I
\end{bmatrix}.
$$
In contrast, our work introduces an additional term, resulting in the flow:
$$
\begin{bmatrix}  - I & \quad I \\\
I - \mu^{-1} \nabla f & \quad - I   - \mu^{-1} N
\end{bmatrix}.
$$
The key contribution lies in the introduction of the $N$ term. Importantly, the strong Lyapunov property still holds for a broad class of $N$, significantly enhancing the generalization capability of the method.

For example, in composite convex optimization, the comparison is as follows:
$$
\begin{bmatrix}  - I & \quad I \\\
I - \mu^{-1} (\nabla f + \partial g) & \quad - I
\end{bmatrix}
\quad \text{vs.} \quad
\begin{bmatrix}  - I & \quad I \\\
I - \mu^{-1} \nabla f & \quad - I - \mu^{-1} \partial g
\end{bmatrix}.
$$
More importantly, by selecting $N(x) = \begin{pmatrix} 0 & B^{\top} \\\ B & 0 \end{pmatrix}$, our approach extends to the min-max problem, which the original rotated gradient flow cannot handle.


**Discrete Level**

On the discrete level, we introduce the AOR (Accelerated Over-Relaxation) technique to approximate the gradient term as:   $$\nabla f(x) \approx 2\nabla f(x_{k+1}) - \nabla f(x_k),$$
whereas Chen and Luo use a simpler discretization, $\nabla f(x) \approx \nabla f(x_{k+1})$, and require an additional gradient step or carefully designed extrapolation scheme to achieve acceleration.

The AOR discretization is more faithful to the structure of the continuous flow. This fidelity not only simplifies convergence analysis but also significantly enhances generalization capabilities. For instance, beyond gradient terms, AOR can also be applied to bilinear couplings, such as   $$Bv \approx B (2v_{k+1} - v_k),$$

resulting in accelerated first-order methods for min-max problems with bilinear coupling.

---

### Meta-Review · Area_Chair_k9jN · 2024-12-19

**Metareview:**

The paper proposes an overrelaxation method, which effectively replaces $\nabla f(x_k)$ with $2\nabla f(x_k) - \nabla f(x_{k-1})$ to "fix" the convergence of the heavy-ball method in smooth strongly convex minimization methods. Namely, while the heavy ball is known to converge for strongly convex quadratics, it is also known to not converge in the worst case for smooth strongly convex minimization problems. The paper proves that the aforementioned strategy suffices to "fix" this issue for smooth strongly convex minimization. It further shows that the same strategy can be used to address convex composite problems and strongly convex-strongly concave min-max optimization problems. The main strengths of the paper lie in the elegance of the provided analysis and in the ability to apply the same approach to multiple settings. For this reason, I believe the paper would be a good addition to the conference.

I have some additional comments for the authors. What the authors call "over relaxation" is a special case of the "operator extrapolation" broadly used in the variational inequalities literature. The operator extrapolation boils down to replacing $F(x_k)$ with $F(x_k) + \theta (F(x_k) - F(x_{k-1})),$ which is the same as the "over relaxation" when $\theta = 1$ (and is, indeed, perhaps the most common choice). The paper does not cite this literature and I believe the authors may not be aware of this connection. The earliest reference is the method of Popov from 1980 when applied to unconstrained problems. It was rediscovered as the "optimistic method" or "optimistic mirror descent" in the more recent literature, starting with Daskalakis et al., 2017. The same idea appears in a paper by Kotsalis, Lan, and Li in SIOPT 2022 (which the authors should consult also in light of questions regarding stochastic settings). It was further used to obtain convergence results for cyclic methods in a line of work starting with Song and Diakonikolas, SIOPT 2023.

While the algorithms and the results in the current paper are not the same as the results from the above line of work, the "operator extrapolation" idea is so closely related that there ought to be a detailed discussion of this line of work and an appropriate comparison to it.

**Additional Comments On Reviewer Discussion:**

The main questions raised by the reviews were in regards to the comparison to related work and motivation for the obtained results, considering that there already existed alternative algorithms addressing the same settings while the proposed method does not outperform them even numerically. These points were somewhat addressed by noting that the proposed approach can address multiple settings in a unified way and arguing about the simplicity of the analysis, which mostly convinced the reviewers.

---

### Decision · Program_Chairs · 2025-01-22

Accept (Poster)